# *Streptococcus pneumoniae* drives specific and lasting Natural Killer cell memory

Tiphaine M. N. Camarasa[1,2☯¤a], Júlia Torné[1☯], Christine Chevalier[1], Orhan Rasid[1¤b], Melanie A. Hamon[1]*

1 Chromatin and Infection Unit, Institut Pasteur, Paris, France, 2 Université Paris Cité, 562 Bio Sorbonne Paris Cité, Paris, France

☯ These authors contributed equally to this work.
¤a Current address: Infection Immunology laboratory, Department of Biomedicine, University of Basel, Basel, Switzerland
¤b Current address: School of Infection and Immunity, University of Glasgow, Glasgow, United Kingdom
* melanie.hamon@pasteur.fr

**Data Availability Statement:** All relevant data are within the manuscript and its Supporting Information files.

**Funding:** This study and M.A.H were supported by Institut Pasteur, the Agence Nationale de la

## Abstract

NK cells are important mediators of innate immunity and play an essential role for host protection against infection, although their responses to bacteria are poorly understood. Recently NK cells were shown to display memory properties, as characterized by an epigenetic signature leading to a stronger secondary response. Although NK cell memory could be a promising mechanism to fight against infection, it has not been described upon bacterial infection. Using a mouse model, we reveal that NK cells develop specific and long-term memory following sub-lethal infection with the extracellular pathogen *Streptococcus pneumoniae*. Memory NK cells display intrinsic sensing and response to bacteria *in vitro*, in a manner that is enhanced post-bacterial infection. In addition, their transfer into naïve mice confers protection from lethal infection for at least 12 weeks. Interestingly, NK cells display enhanced cytotoxic molecule production upon secondary stimulation and their protective role is dependent on Perforin and independent of IFNγ. Thus, our study identifies a new role for NK cells during bacterial infection, opening the possibility to harness innate immune memory for therapeutic purposes.

## Author summary

Natural Killer (NK) cells serve as crucial effectors of the innate immune system and play a vital role in safeguarding the host against infections. It has recently emerged that NK cells exhibit characteristics of immunological memory resulting in a heightened response upon a second encounter with the same pathogen. Although the potential of NK cell memory in combating infections holds promise, cellular responses and memory functions to bacterial infections have not yet been elucidated. Using *Streptococcus pneumoniae* as our model bacterium, we reveal that NK cells sense and respond to bacteria, as well as develop specific memory properties. Remarkably, transferring NK cells from previously infected mice to naïve ones provided protection against lethal infection for at least 12 weeks. Furthermore, we show that memory NK cells produced more cytotoxic molecules upon

Recherche (ANR-17-CE12-0007), the Fondation pour la Recherche Médicale (EQU202003010152), the Fondation iXCore-iXLife, the Don Prix CANETTI 2020, the EMBO Young Investigator Program. M.A. H is a member of the Laboratoire d'Excellence "Integrative Biology of Emerging Infectious Diseases" Agence Nationale de la Recherche (ANR): ANR-10-LABX- 62-EIBID). T.M.N.C received a salary from the Crédit Agricole d'Ile de France and Fondation pour la Recherche Médicale, grants no. PMJ201810007628 (Prix MARIANE JOSSO) and no. FDT202106012790 (Fin de thèse). J.T received a salary from the Fondation ARC, grant no. ARCPOST-DOC2021070004074. The funders had no role in study design, data collection and analysis, decision to publish, or preparation of the manuscript.

**Competing interests:** The authors have declared that no competing interests exist.

secondary stimulation. These findings unravel a novel role for NK cells in the context of bacterial infections, thereby opening avenues for harnessing the potential of innate immune memory for therapeutic applications.

## Introduction

In the last decade, the discovery of memory responses mediated by innate immune cells has questioned the dogma that only adaptive immune cells retain memory of previous exposure. Monocytes and macrophages have been shown to acquire an innate immune memory described by a faster and greater response against a secondary challenge with homologous or even heterologous pathogens [1,2]. This capacity, termed trained immunity, is defined by a return to basal state of activation after removal of the primary stimulus, accompanied by the acquisition of persistent epigenetic changes and metabolic rewiring necessary for enhanced response to secondary challenge [3,4].

Natural Killer (NK) cells also display memory properties, which are different from those of macrophages and monocytes. Indeed, memory NK cells display high antigen-specificity and undergo clonal expansion of specific receptor-defined sub-populations [5–7]. NK cell memory has been well studied in the context of viral infections (CMV, Epstein-Barr virus, etc.) [8–12], however, evidence of memory following bacterial infection, especially extracellular bacteria, remains undefined. In fact, the role of NK cells during bacterial infections is controversial [13]. IFNγ produced by NK cells, contributes to bacterial clearance by activating immune cells, such as macrophages and neutrophils to enhance phagocytosis in the case of lung infections with *L. pneumophila*, *K. pneumoniae* or *M. tuberculosis* among others [14–17]. However, upon infection with bacillus Calmette-Guérin (BCG), *S. pneumoniae* or *S. pyogenes*, NK cells can be deleterious, leading to tissue damage associated with an excessive inflammatory response [18–21]. In addition, although some studies show that NK cells induce apoptosis of infected cells [22–24], others find that certain intracellular bacterial infections are not affected by NK cells [25,26].

*Streptococcus pneumoniae* (also known as pneumococcus) is a Gram-positive, extracellular bacterium and natural colonizer of the human upper respiratory tract but also an opportunistic pathogen [27]. Indeed, although *S. pneumoniae* resides asymptomatically in its natural reservoir, the nasopharynx, it can spread to the lungs or enter the bloodstream causing deadly invasive inflammatory diseases such as pneumonia, meningitis and sepsis [28–30]. Comparable to other bacterial infections, the role of NK cells during *S. pneumoniae* infection is poorly understood. NK cells contribute to infection clearance by IFNγ production in the lungs [31,32] however, NK cell production of IL-10 is detrimental to infected mice [33].

In this study, we used *S. pneumoniae* to explore NK cell responses to an extracellular bacterial infection. We show that NK cells directly sense and respond to *S. pneumoniae*, as well as acquire memory properties. Upon clearance of a primary infection, NK cells retain intrinsic heightened *in vitro* response to *S. pneumoniae*, and *in vivo* they acquire protective functions against lethal infection. Interestingly, we demonstrate that NK cell protection is mediated through higher levels of cytotoxic products (Granzyme B and Perforin), and not through IFNγ mediated responses or inflammatory cell recruitment, thereby revealing a novel role for NK cells in bacterial infection.

## Results

### NK cells acquire intrinsic memory features 21 days after *in vivo* infection

We set up model in which mice were infected intranasally with two consecutive sub-lethal doses of *S. pneumoniae* (SPN, 5x10$^5$ CFU) or treated with PBS (Fig 1A). Infecting mice with

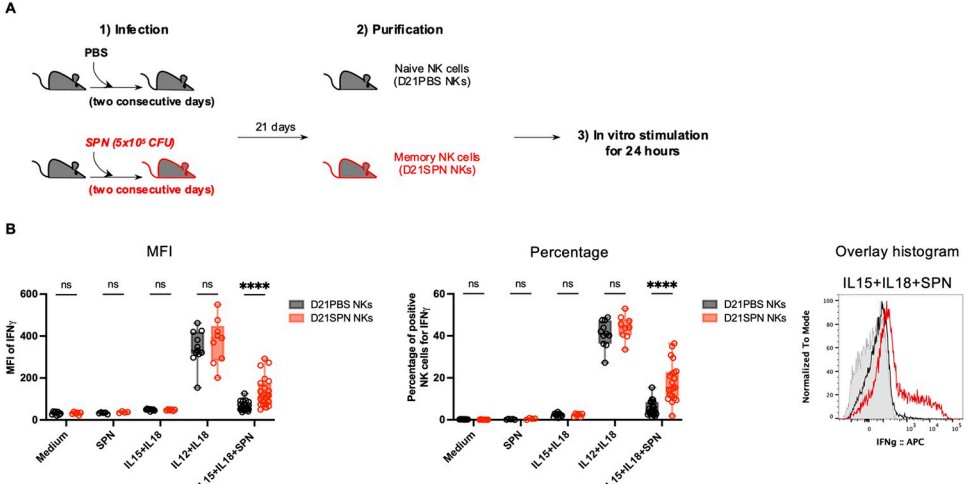

**Fig 1. NK cells acquire intrinsic memory features 21 days after *in vivo* infection. (A)** Experimental scheme. C57BL/6 mice were intranasally injected with either PBS (black symbols) or sub-lethal dose of *S. pneumoniae* (SPN, red symbols, 5x10^5 CFU) for two consecutive days. After 21 days, NK cells were highly purified from spleens of D21PBS or D21SPN mice (98% of purity) and stimulated *in vitro* with cytokines (IL-15 at 2 ng/ml, IL-18 at 1,5 ng/ml, IL-12 at 1,25 ng/ml) and formaldehyde inactivated *S. pneumoniae* (SPN, MOI 20) for 24 hours. **(B)** Intensity of IFNγ expression in NK cells (MFI, **left panel**), percentage of IFNγ+ NK cells (**middle panel**), representative overlay histogram upon IL-15+IL-18+SPN stimulation (**right panel**, gray represents isotype control). D21PBS NK cells and D21SPN NK cells are purified from spleen, pooled from n ≥ 4 mice/group and incubated in n ≥ 3 experimental replicates/group. Box plots where each dot represents an experimental replicate (black dots for D21PBS NK cells, red dots for D21SPN NKs cells), lines are median, error bars show min to max. Data are representative of at least three experiments. ns, not significant. **** p < 0.0001. 2way ANOVA test comparing D21PBS NKs and D21SPN NKs values in each condition.

two consecutive doses over two days improved reproducibility of our results compared with one dose, suggesting robust colonization is important. 21 days later, to test their intrinsic features, NK cells from the spleen were extracted, highly purified by negative selection (~98%, S1A Fig), and stimulated *ex vivo*. Naïve (D21PBS NKs) or previously exposed NK cells (D21SPN NKs) were incubated for 24 hours with different combinations of cytokines and/or formaldehyde inactivated SPN (Fig 1B). Upon incubation of NK cells with either medium alone, SPN alone, or IL-15+IL-18 (to ensure maintenance of cell viability), no production of IFNγ was detected. In contrast, IL-12+IL-18 (activating cytokines) activated both D21PBS and D21SPN NK cells to the same levels. These results show that under these *in vitro* conditions NK cells can be reproducibly activated. Strikingly, upon incubation with IL-15+IL-18+SPN, D21SPN NK cells displayed significantly higher signal per cell (Mean fluorescence intensity, MFI) and percentages of IFNγ+ cells compared with control D21PBS NK cells (Fig 1B). These results indicate that memory IFNγ+ NK cells are both more numerous and express IFNγ to higher levels upon secondary stimulation with *S. pneumoniae*. Since the higher response of these cells occurs only under condition where bacteria are present with IL-15+IL-18, we can rule out that D21SPN NK cells are hyperactivated under any condition and are specifically responding to SPN. Therefore, NK cells from infected mice acquire intrinsic memory features which are detected *ex vivo* and characterized by sensing *S. pneumoniae* and responding more strongly upon secondary stimulation compared to primo-stimulation.

Innate immune memory in macrophages, monocytes and NK cells has been described to correlate with chromatin modifications in various models [3,34]. In particular, we have previously shown in [35] that NK cell memory in a post-endotoxemia model relies on H3K4me1 histone modification at an *ifng* enhancer. To explore whether this might be the case for NK cell memory to *S. pneumoniae*, we assessed the histone mark H3K4me1 associated with

upstream regulatory regions of *ifng* gene in D21PBS and D21SPN NK cells by chromatin immunoprecipitation followed by qPCR (ChIP-qPCR). Interestingly, although we did not reach statistical significance, we always observed the same consistent increase of H3K4me1 at the *ifng* enhancer located at -22 kb in memory NK cells compared to naïve cells (S1B Fig). By contrast, we did not observe a difference in H3K4me1 at the -55 kb enhancer, suggesting histone modifications are occurring at specific regulatory loci. In addition, the increase of H3K4me1 we observed in D21SPN NK cells was located at the same region (-22 kb) as post-endotoxemia memory NK cells. Together, these data suggest that NK cell memory is associated with epigenetic changes at upstream regulatory regions of *ifng* gene.

The Ly49H, NKG2C and NKG2A receptors have been shown to be implicated in NK cell recognition and memory responses to MCMV, HCMV and EBV respectively, leading to an expression increase upon infection [8–10]. We tested the expression levels of several NK cell receptors in post-SPN NK cells. As early as 12 days post-infection, we observe no difference in the percentage of NKG2D, Ly49D, Ly49H, or Ly49F positive cells between D12PBS and D12SPN NK cells. A small increase in the percentage of Ly49C/I$^+$ NK cells is observed, which was no longer detectable 21 days post-infection (S1C and S1D Fig). Therefore, although NK cells acquire memory properties post-bacterial infection, none of the known NK cell receptors previously associated with memory NK cells to viruses seem to be involved.

## Sub-lethal infection with *S. pneumoniae* induces a rapid and transient immune response that returns to basal state before 21 days

To understand the immune environment from which NK cells were extracted, we studied a primary infection. Mice were infected as described Fig 1A, and organs were collected at 24 hours, 72 hours and 21 days post-infection (Fig 2A). Using this infection protocol, mice displayed no weight loss and showed minimal clinical signs (S2A and S2B Fig). At 24 hours, bacteria were mostly located in the nasal cavity (Fig 2B) and spread into bronchio-alveolar lavage fluid (BALF) and lungs at 72 hours post-infection (Fig 2C). Importantly, no dissemination was observed following this dose of infection, as demonstrated by the undetectable levels of bacteria in the blood and spleen at both 24 and 72 hours (Fig 2B–2C).

The sub-lethal infection that we perform is accompanied by a low-level immune response. Indeed, at 24 hours, an increase in the percentage and number of neutrophils in the lungs and BALF was detectable, but rapidly decreased to basal levels at 72 hours (Fig 2B and 2C, numbers in S2C and S2D Fig). Furthermore, a small increase of CD69$^+$ neutrophils was detected in the BALF at 24 and 72 hours, suggesting a low level of activation of these cells (Fig 2B and 2C). In contrast to neutrophils, we did not detect an increase in NK cell percentages or numbers in the lungs at any time point (Fig 2B and 2C, numbers in S2C and S2D Fig). In addition, we tested responsiveness of NK cells by measuring IFNγ, CD69, Perforin and Granzyme B expression. A small increase in CD69$^+$ NK cells was observed in the lungs only at 24 hours and 72 hours (S2C and S2D Fig), but no increase of Perforin$^+$ and Granzyme B$^+$ cells was detected at both 24 and 72 hours post-infection (Figs 2B and 2C, MFI in S2C and S2D). Together these data indicate that sub-lethal infection with SPN induces a low, rapid and transient immune response characterized by the recruitment of neutrophils but no significant NK cell responses.

To study the inflammatory environment at 21 days post-infection, we harvested organs and performed bacterial enumeration. At this time point, no bacteria were recovered in the nasal lavage, BALF, lungs, or spleen (Fig 2D), demonstrating that bacteria had completely been cleared. Additionally, the immune parameters returned to pre-infection state as the percentage and number of neutrophils and NK cells detected in 21 days infected mice was the same as in control PBS mice (Fig 2D, numbers in S2E Fig). Similarly, both neutrophils and NK cells

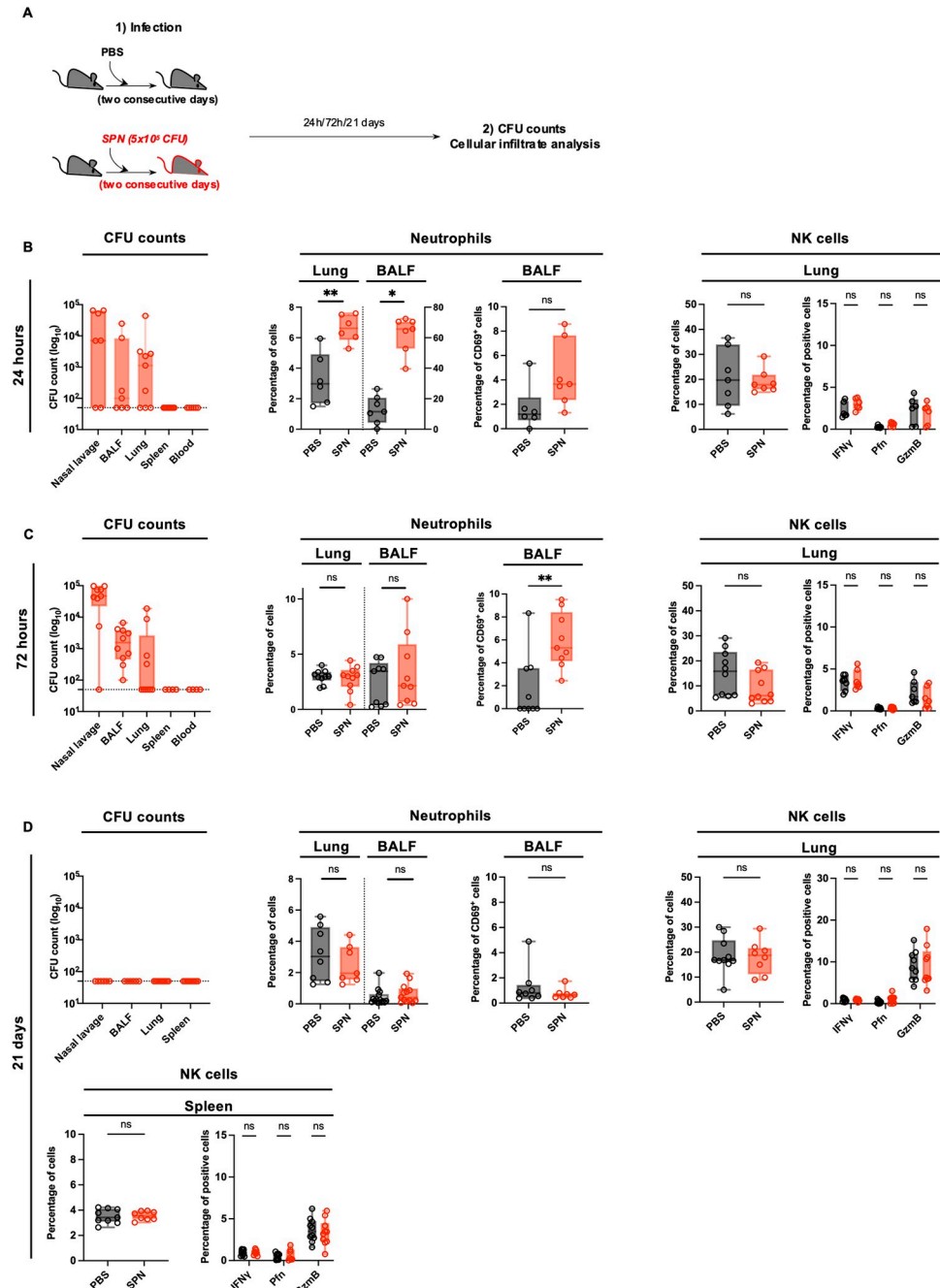

**Fig 2. Sub-lethal infection with *S. pneumoniae* induces a rapid and transient immune response that returns to basal state before 21 days. (A)** Experimental scheme, showing that C57BL/6 mice were intranasally injected with either PBS (black symbols) or sub-lethal dose of *S. pneumoniae* (SPN, red symbols, 5x10$^5$ CFU) for two consecutive days. Organs were collected at 24h, 72h and 21 days post-infection for CFU counts and cellular infiltrate determination by flow cytometry. **(B-D)** Organs were collected at 24h (**B**), 72h (**C**) and 21 days post-infection (**D**). CFU counts in the nasal lavage, BALF, lungs, spleen and blood of infected mice (**left panel**). Box plots where each dot represents an individual mouse, lines are the mean, error bars show min to max and dotted lines represent limit of detection. Percentage of neutrophils (CD11b$^+$ Ly6G$^+$) among CD45$^+$ cells in the lungs and bronchio-alveolar lavage fluid (BALF), percentage of CD69$^+$ neutrophils in the BALF (**middle panel**). Percentage of NK cells (NK1.1$^+$ CD3$^-$) among CD45$^+$ cells, percentage of IFNγ$^+$, Perforin$^+$, Granzyme B$^+$ NK cells in the lungs and spleen (**right panel**). Box plots where each dot represents an individual mouse (black dots for uninfected mice, red dots for infected mice), lines are the median, error bar show min to max. Data are pooled from two or three repeats with n ≥ 3 mice/group. ns, not significant. * p < 0.05 and ** p <0.01, Mann-Whitney test for single comparisons and 2way ANOVA test for multiple comparisons.

showed no sign of activation in the BALF, lungs and spleen respectively (Fig 2D and S2E Fig). Therefore 21 days after primary exposure, infection by *S. pneumoniae* is no longer detected and immune cells are similar to uninfected conditions, both in their number and level of activation.

## Transferred memory NK cells protect mice from lethal *S. pneumoniae* infection for at least 12 weeks

As NK cells display intrinsic sensing and memory activities *in vitro*, we investigated the potential role of D21SPN NK cells during lethal *S. pneumoniae* challenge *in vivo*. We adoptively transferred purified D21PBS or D21SPN NK cells into naïve mice ($2x10^5$ cells, intravenously, Fig 3A) and one day after the transfer, recipient mice were intranasally infected with a lethal dose of SPN ($5x10^6$ CFU for survival study and $1x10^7$ CFU for bacterial counts comparison). At all times points, transferred congenic CD45.2$^+$ NK cells were circulating and detectable at similar percentages in lungs and blood, suggesting there is no preferential trafficking between D21PBS and D21SPN NK cells (S3A and S3B Fig). Importantly, we observed a significant reduction in bacterial counts at 40 hours in all tested organs of mice having received D21SPN NK cells compared with those having received D21PBS NK cells, which is not observed at 24 hours (Fig 3B). Additionally, the transfer of D21SPN NK cells reduced mortality and clinical signs in recipient mice infected with *S. pneumoniae* ($5x10^6$ CFU, Fig 3C and 3D). We observed a 75% survival rate for mice having received D21SPN NK cells compared to only 25% for those having received D21PBS NK cells. Interestingly, both groups of mice lost similar weight during the first days of infection (Fig 3E), but mice that received D21SPN NK cells fully recovered from the lethal infection by 7 days as displayed both by clinical scores and weight. We further tested the same experimental set up but purifying and transferring NK cells from lungs of naïve (D21PBS) or previously infected mice (D21SPN) (Fig 3F). Interestingly, we observed that D21SPN NK cells from lungs had a similar protective effect than those from spleen and contribute to reduce the numbers of CFU in recipient mice. Thus, our results demonstrate that the transfer of as few as $2x10^5$ D21SPN NK cells from either spleen or lungs confers significant protection during lethal *S. pneumoniae* infection compared to D21PBS NK cells.

To evaluate long term protection, we purified NK cells from spleen of naïve or previously infected mice 12 weeks after primary exposure (W12PBS NKs or W12SPN NKs) and transferred them prior to lethal infection with *S. pneumoniae*. Remarkably, we observed a similar significant reduction in bacterial counts in the lungs and spleen of mice having received W12SPN NK cells, as cells from 21 days post- primary exposure (Fig 3G). Therefore, NK cells retain memory of a primo *S. pneumoniae* infection, and maintain the ability to protect against lethal challenge, for at least 12 weeks.

To assess the specificity of the observed memory properties, we performed the same transfer model as described in Fig 3A, but infected recipient mice with a lethal dose of the heterologous bacterium *Listeria monocytogenes* ($1x10^6$ CFU, intravenously). Organs were collected 40 hours post-infection for bacterial enumeration (Fig 3H). Our results showed no differences in bacterial counts in the spleen of infected mice between the two conditions. Surprisingly we counted higher numbers of bacteria in the liver of mice having received D21SPN NK cells than D21PBS NK cells. Thus, *in vivo*, *S. pneumoniae* memory NK cells are not able to protect recipient mice from *L. monocytogenes* infection, demonstrating a specificity in their response to *S. pneumoniae*. Together, these data show that memory NK cells protect mice upon secondary *in vivo* infection, with specific and long-term properties.

## Protection of mice mediated by memory NK cells is IFNγ independent

Because D21SPN NK cells showed higher expression of IFNγ upon secondary stimulation *in vitro* (Fig 1B), we hypothesized that IFNγ production by memory NK cells *in vivo* could be an

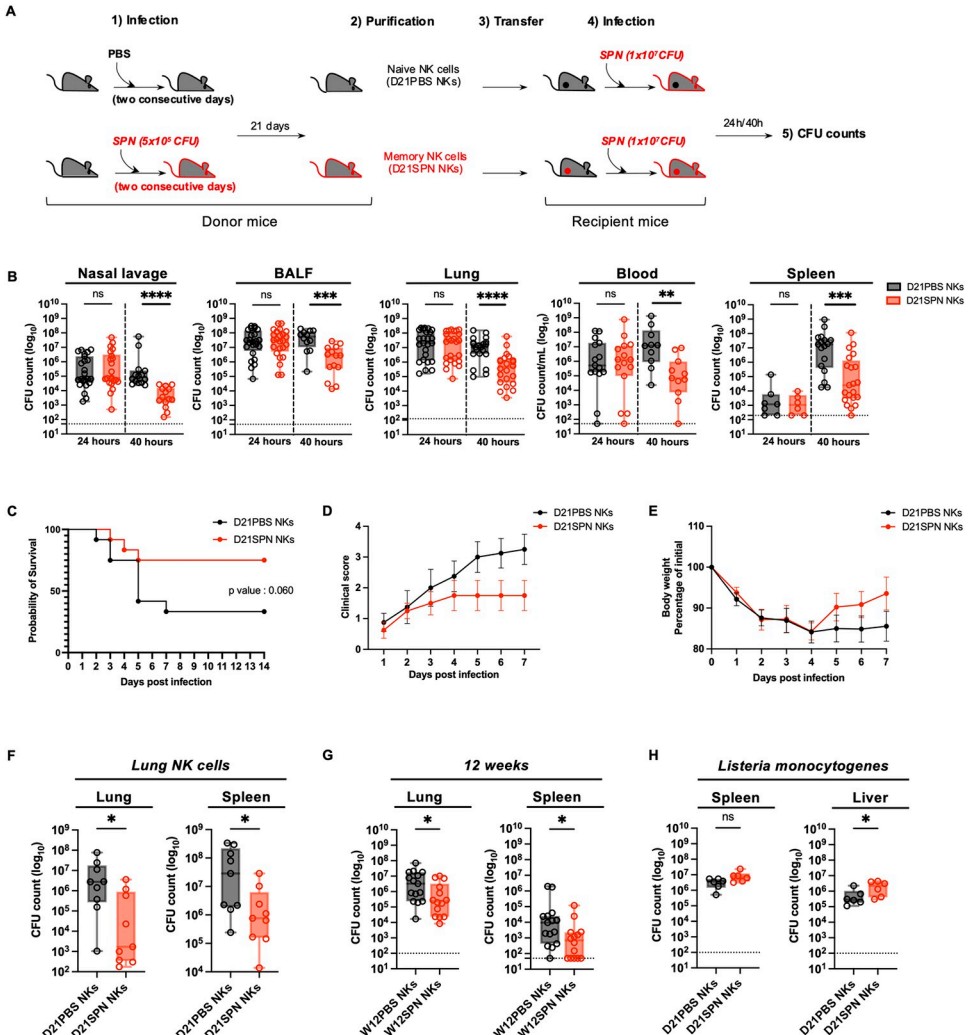

**Fig 3. Transferred memory NK cells protect mice from lethal *S. pneumoniae* infection for at least 12 weeks. (A)** Experimental scheme. C57BL/6 mice (donor mice) were intranasally injected with either PBS (black symbols) or sub-lethal dose of *S. pneumoniae* (SPN, red symbols, $5 \times 10^5$ CFU) for two consecutive days. After 21 days or 12 weeks, NK cells from spleens or lungs were highly purified (98%) and transferred into naïve mice (recipient mice, intravenously, $2 \times 10^5$ cells). One day after, all recipient mice were intranasally infected with a lethal dose of *S. pneumoniae* ($5 \times 10^6$ CFU for survival study, $1 \times 10^7$ CFU for bacterial counts comparison) or *L. monocytogenes* ($1 \times 10^6$ CFU, intravenously). Organs were collected at 24h and 40h post-infection for CFU counts and flow cytometry analysis. **(B)** Bacterial counts at 24h and 40h post-infection in the nasal lavage, bronchio-alveolar lavage fluid (BALF), lungs, blood and spleen of mice having received either D21PBS NKs (black symbols) or D21SPN NKs (red symbols). Box plots where each dot represents an individual mouse, lines are the mean, error bars show min to max and dotted lines represent limit of detection. Data are pooled from at least two repeats with n ≥ 3 mice/group. **(C)** Survival curve. Dots represent the percentage of survival of total mice. Data are representative of three repeats with n = 4 mice/group. **(D)** Clinical score. Dots represent the mean, error bars show the standard error of the mean (SEM). Data are representative of two repeats with n = 4 mice/group. **(E)** Weight represented as percentage of initial body weight loss. Dots represent the mean, error bars show the standard error of the mean (SEM). After death, mice have the value of 80%. Data are representative of two repeats with n = 4 mice/group. **(F)** Bacterial counts at 40h post-infection in the lungs and spleen of mice having received either D21PBS NKs (black symbols) or D21SPN NKs (red symbols) isolated from the lungs of donor mice. Box plots where each dot represents an individual mouse, lines are the mean, error bars show min to max and dotted lines represent limit of detection. Data are pooled from three repeats with n ≥ 3 mice/group. **(G)** Bacterial counts at 40h post-infection in the lungs and spleen of mice having received either W12PBS NKs (black symbols) or W12SPN NKs (red symbols). Box plots where each dot represents an individual mouse, lines are the mean, error bars show min to max and dotted lines represent limit of detection. Data are pooled from three repeats with n ≥ 4 mice/group. **(H)** Bacterial counts at 40h post-infection in the spleen and liver of mice infected with *Listeria monocytogenes* (intravenously) and having previously received either D21PBS NKs (black symbols) or D21SPN NKs (red symbols) from the spleens of donor mice. Box plots where each dot represents an individual mouse, lines are the mean, error

bars show min to max and dotted lines represent limit of detection. Data are pooled from two repeats with n = 3 mice/group. ns, not significant. * p < 0.05, ** p < 0.01, *** p < 0.001 **** p < 0.0001. 2way ANOVA **(B)**, Log-rank (Mantel-Cox) **(C)** and Mann-Whitney **(F,G,H)** tests for statistical significance.

important factor for protecting mice against lethal infection. Using CD45.1 and CD45.2 congenic mice, we compared intracellular IFNγ expression between transferred and endogenous NK cells in both groups of mice receiving D21PBS or D21SPN NK cells (Fig 4A). Upon lethal infection with SPN, IFNγ expression by NK cells increased at 40 hours in the lungs at equal levels between endogenous and transferred cells in the same mouse. But surprisingly, both endogenous and transferred NK cells in mice receiving D21SPN NK cells expressed lower levels of IFNγ than those having received D21PBS NK cells. In addition, total lung IFNγ levels showed no difference between protected mice and control mice (Fig 4B). Therefore, memory NK cells are not producing more IFNγ upon secondary stimulation *in vivo*.

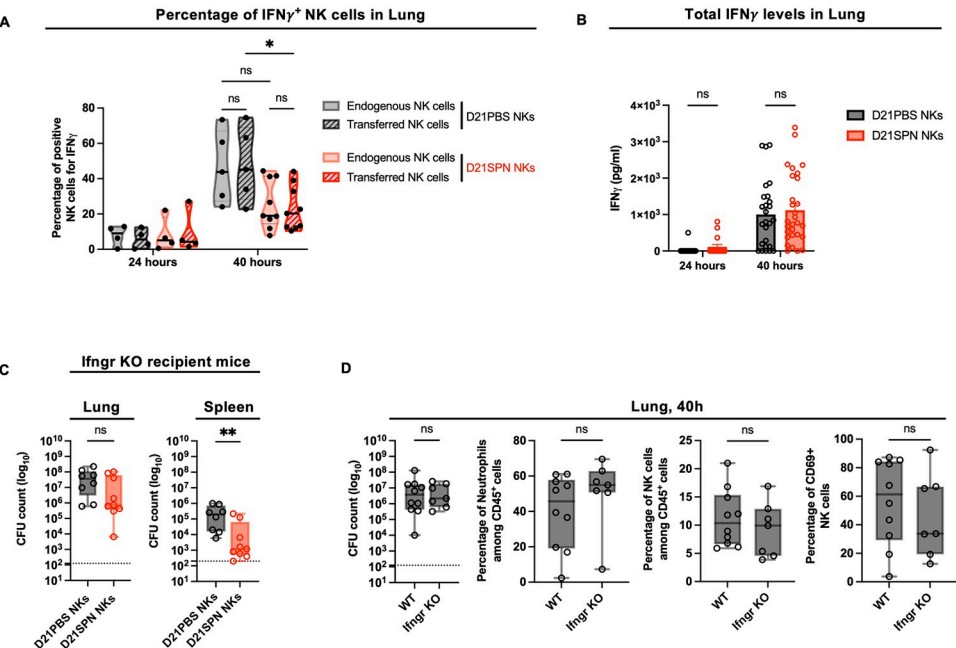

**Fig 4. Protection of mice mediated by memory NK cells is IFNγ independent. (A-B)** Mouse infections are carried out as in the scheme in Fig 3A. Organs were collected at 24h and 40h post-infection for flow cytometry analysis and ELISA assays. **(A)** Flow cytometry analysis of IFNγ expression in CD45.1[+] endogenous NK cells and CD45.2[+] transferred NK cells (hatched) into both mice having received D21PBS NK cells (black) and mice having received D21SPN NK cells (red). Violin plot where each dot represents an individual mouse, lines are the median. Data are pooled from two repeats with n ≥ 2 mice/group. **(B)** IFNγ ELISA assays of lung supernatants from infected mice having received either D21PBS NKs (black) or D21SPN NKs (red) at 24h and 40h post-infection. Bars are the mean of at least four experiments with n ≥ 4 mice/group, each dot represents an individual mouse and error bars are the standard error of the mean (SEM). **(C)** Mouse infections are carried out as in the scheme in Fig 3A, with the exception that WT mice were used as donor mice and Ifngr KO mice as recipient mice. Bacterial counts at 40h post-infection in the lungs and spleen of Ifngr KO mice having received D21PBS WT NK cells (black symbols) or D21SPN WT NK cells (red symbols). Box plots where each dot represents an individual mouse, lines are the mean, error bars show min to max and dotted lines represent limit of detection. Data are pooled from two repeats with n ≥ 4 mice/group. **(D)** WT and Ifngr KO mice were intranasally infected with a lethal dose of *S. pneumoniae* (1x10[7] CFU). Lungs were collected at 40h post-infection for CFU counts and flow cytometry analysis. Percentages of neutrophils (Ly6G[+] CD11b[+]), NK cells (NK1.1[+] CD3[-]) and CD69[+] NK cells. Box plots where each dot represents an individual mouse, lines are the mean, error bars show min to max and dotted lines represent limit of detection. Data are pooled from two repeats with n ≥ 3 mice/group ns, not significant. * p < 0.05, ** p < 0.01. 2way ANOVA **(A,B)** and Mann-Whitney **(C,D)** tests for statistical significance.

To further address the role of IFNγ in protection, we transferred WT naïve or WT memory NK cells into IFNγ receptor deficient mice (Ifngr KO, Fig 4C). One day after the transfer, we infected Ifngr KO recipient mice with the same lethal dose of SPN as in Fig 3B and collected organs at 40 hours post-infection. By comparing both WT and Ifngr KO mice having received naïve NK cells we did not detect significant differences in bacterial counts, recruitment of neutrophils or activation of NK cells in the lungs at 40 hours (Fig 4D). These results suggest that IFNγ signaling is not protective during early times of SPN infection. Importantly, we still found a significant reduction in bacterial numbers in the spleen, and a similar trend in lungs, of Ifngr KO recipient mice having received WT D21SPN NK cells compared to Ifngr KO recipient mice having received WT D21PBS NK cells (Fig 4C). Strikingly, these data suggest that IFNγ signaling is not necessary for the protection of mice having received memory NK cells.

## Protection of mice is an intrinsic NK cell property

To assess the protective functions of memory NK cells, we analyzed the innate immune response in the lungs of recipient mice. Using the transfer model described in Fig 3A, we infected recipient mice with a lethal dose of *S. pneumoniae* ($1x10^7$ CFU) and collected organs at 24 and 40 hours post-infection. With this infection protocol, bacteria are detectable in the BALF and lungs and disseminate to the bloodstream and spleen by 24 hours (Fig 3B). We first compared immune cell recruitment in the lungs and BALF of infected mice having received naïve or memory NK cells (Fig 5A and 5B, full gating strategy in S4A Fig). Although we observed a robust recruitment of neutrophils at 24 hours in the lungs and BALF of infected mice, we found similar percentages and numbers of neutrophils between the two groups of recipient mice. We observed the same reduction in the percentages and numbers of alveolar macrophages in the lungs and BALF of infected mice having received either naïve or memory NK cells. Surprisingly, we do not observe a significant recruitment or increase in other immune cell types (interstitial macrophages, eosinophils, dendritic cells, monocytes, T cells or NK cells) in the lungs of either group of infected mice compared to uninfected.

We next evaluated innate immune cell activation in the lungs of recipient mice at 40 hours post-infection. Although neutrophils and NK cells upregulated CD69 compared to uninfected animals, we did not detect any difference between mice having received naïve or memory NK cells (Fig 5C). Additionally, in neutrophils, we did not observe an increase in the intensity of CD11b expression (MFI) or in the percentage of ROS$^+$ cells compared to uninfected cells (S4B Fig). Furthermore, interstitial macrophages had the same values of CD86$^+$ cells and intensity of MHCII expression between mice having received D21PBS or D21SPN NK cells (Figs 5C and S4B). Therefore, alongside no differential immune cell recruitment, we did not detect an increase in innate immune cell activation in the lungs of protected mice. These results suggest that the protection provided by memory NK cells is not through enhanced recruitment or activation of any of the inflammatory cells tested.

To further investigate the mechanisms of protection mediated by memory NK cells, we quantified selected cytokines and chemokines in lung supernatants by ELISA (Fig 5D at 40 hours, S4C Fig at 24 hours). Coherent with our observation that innate immune cells are not recruited and activated to higher levels by memory NK cells, we did not detect an increase of pro-inflammatory cytokines in the lungs of mice having received D21SPN NK cells. In fact, we showed a signification reduction of CXCL1 production in the lungs of protected mice (Fig 5D).

In addition to cytokines/chemokines signaling, we also studied release of cytotoxic molecules *in vivo* by comparing Granzyme B production in the lungs of infected mice having

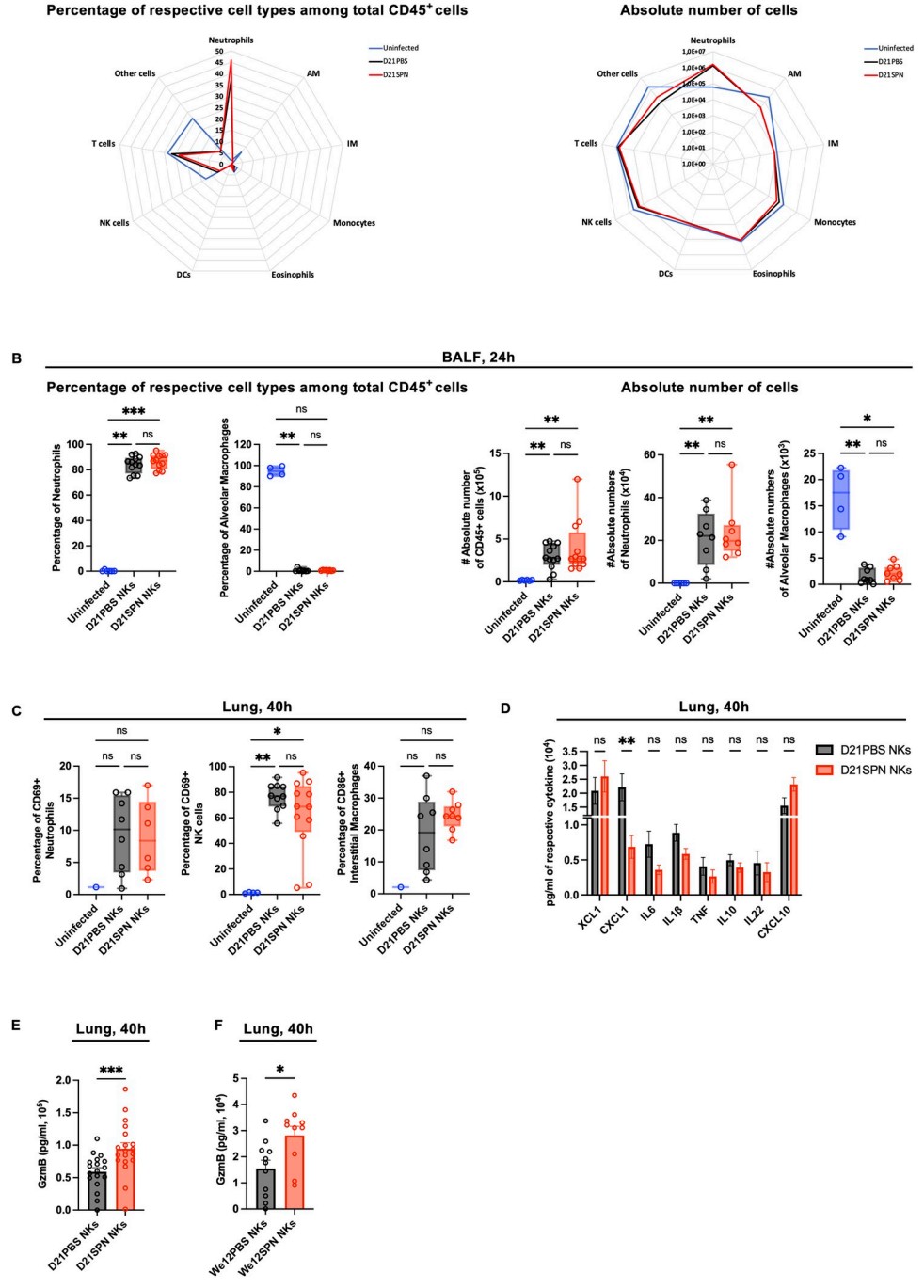

**Fig 5. Protection of mice appears to be an intrinsic NK cell property.** Mouse infections are carried out as in the scheme in Fig 3A. Organs were collected at 24h and 40h post-infection for flow cytometry analysis, ELISA assays and CFU counts. **(A)** Cellular infiltrate analysis in the lungs at 24h post-infection. Percentage of respective cell types among CD45+ cells (**left panel**) and absolute number of cells for each cell type (**right panel**): neutrophils (Ly6G+ CD11b+), alveolar macrophages (AM) (Ly6G- SiglecF+ CD64+ CD11b-), interstitial macrophages (IM) (Ly6G- SiglecF- CD11bhigh MHCII+ CD64+ CD24-), monocytes (Ly6G- SiglecF- CD11bhigh MHCII-), eosinophils (Ly6G- SiglecF+ CD11b+), dendritic cells (DCs) (CD103+ DCs: Ly6G- SiglecF- CD11blow CD103+ CD24+ + CD11b+ DCs: Ly6G- SiglecF- CD11bhigh MHCII+ CD64- CD24+), NK cells (NK1.1+ CD3-), T cells (NK1.1- CD3+). Radar plots with each value representing the geometric mean of three repeats with n = 4 mice. The blue line for uninfected mice, black line for mice having received D21PBS NKs and red line for mice having received D21SPN NKs. **(B)** Cellular infiltrate analysis in the bronchio-alveolar lavage fluid (BALF) at 24h post-infection. Percentage of respective cell types among CD45+

cells (**left panel**) and absolute number of cells for each cell type (**right panel**): neutrophils (Ly6G⁺ CD11b⁺), alveolar macrophages (SiglecF⁺ CD64⁺ CD11b⁻). Box plots where each dot represents an individual mouse (blue dots for uninfected mice, black dots for mice having received D21PBS NKs, red dots for mice having received D21SPN NKs), lines are the median, error bar show min to max. Date are pooled from at least two repeats with n ≥ 2 mice/group. **(C)** Cellular activation analysis in the lung at 40h post-infection. Percentages of CD69⁺ neutrophils, CD69⁺ NK cells and CD86⁺ interstitial macrophages. Box plots where each dot represents an individual mouse (the blue dots are for uninfected mice, black dots for mice having received D21PBS NKs, red dots for mice having received D21SPN NKs), lines are the median, error bar show min to max. Data are pooled from two repeats with n ≥ 1 mice/group. **(D)** ELISA assays of lung supernatants from infected mice at 40h post-infection. Bars are the mean of at least 3 experiments with n ≥ 3 mice/group, error bars are the standard error of the mean (SEM). **(E-F)** At 40h post-infection, lung supernatants of mice having received either D21PBS NKs (black) or D21SPN NKs (red) **(C)**—either We12PBS NKs (black) or We12SPN NKs (red) **(D)** were collected to perform Granzyme B ELISA assays. Bars are the mean of at least 3 experiments with n ≥ 3 mice/group, each dot represents an individual mouse and error bars are the standard error of the mean (SEM). ns, not significant. * p < 0.05, ** p < 0.01, *** p < 0.001. Kruskal-Wallis **(A,B,C)**, 2way ANOVA **(D)** and Mann-Whitney **(E,F)** tests for statistical significance.

previously received D21PBS or D21SPN NK cells (scheme in Fig 3A). Importantly, ELISA on lung supernatants showed an increase of Granzyme B at 40 hours post-infection in mice having received D21SPN NK cells compared to control (Fig 5E), which was not observed at 24 hours post-infection (S4D Fig). Interestingly, the decrease in bacterial numbers in mice having received D21SPN NK cells is detectable at 40h, not at 24h (Fig 3B). Furthermore, we also found the same increase of Granzyme B production in the lungs of mice having received long term memory NK cells (We12SPN NKs, Fig 5F). Therefore, transferring memory NK cells induced a greater production of Granzyme B in the lungs of infected mice. Thus, our results open the possibility that cytotoxic functions of memory NK cells could be important for the protection of mice upon secondary lethal *S. pneumoniae* infection.

## Cytotoxic proteins are upregulated by memory NK cells and important for protection of mice

To further investigate an upregulation of cytotoxic proteins by memory NK cells, we stimulated purified D21PBS and D21SPN NK cells *in vitro* with inactivated *S. pneumoniae*. Interestingly, cytokines alone did not induce Granzyme B or Perforin expression in either naïve or memory NK cells (Fig 6A and 6B). However, addition of *S. pneumoniae* stimulated Granzyme B and Perforin expression specifically in memory NK cells. Indeed, an increase in both percentage of positive cells and intensity of expression was detected under this condition (Fig 6A and 6B). Consistent with the literature, we have observed that responding NK cells (Gzmb⁺) tend to be more mature than non-responding cells (Gzmb⁻), however, we did not observe differences between naïve or memory NK cells in maturation markers (S5A Fig). Interestingly, the incubation of memory NK cells with inactivated *L. monocytogenes* or *Streptococcus agalactiae* (GBS) did not induce an increase of Granzyme B and Perforin (Fig 6A and 6B), supporting once again the specificity of the response. Therefore, NK cells respond to and remember SPN in an intrinsic and specific manner, by releasing cytotoxic proteins. Such a result implies that NK cells are sensing and responding to bacteria. Therefore, we tested whether Toll like Receptors (TLRs) might play a role. To address this, we stimulated D21PBS and D21SPN NK cells *in vitro* with TLR agonists such as LPS (TLR4 agonist) or the Pam3CSK4 (TLR1/TLR2 agonist), in parallel to SPN. We hypothesized that if memory NK cells respond to bacterial stimulation in a TLR mediated manner, they would have an increased response when stimulated by LPS or Pam3CSK4. While memory NK cells had increased levels of Granzyme B following SPN stimulation, we did not observe differences in D21SPN compared to D21PBS NK cells stimulated by LPS or Pam3CSK4 (Fig 6C). As a control, we have followed NK cells response to LPS or the TLR agonist by measuring IFNγ positive cells (S5B Fig). These results suggest that memory

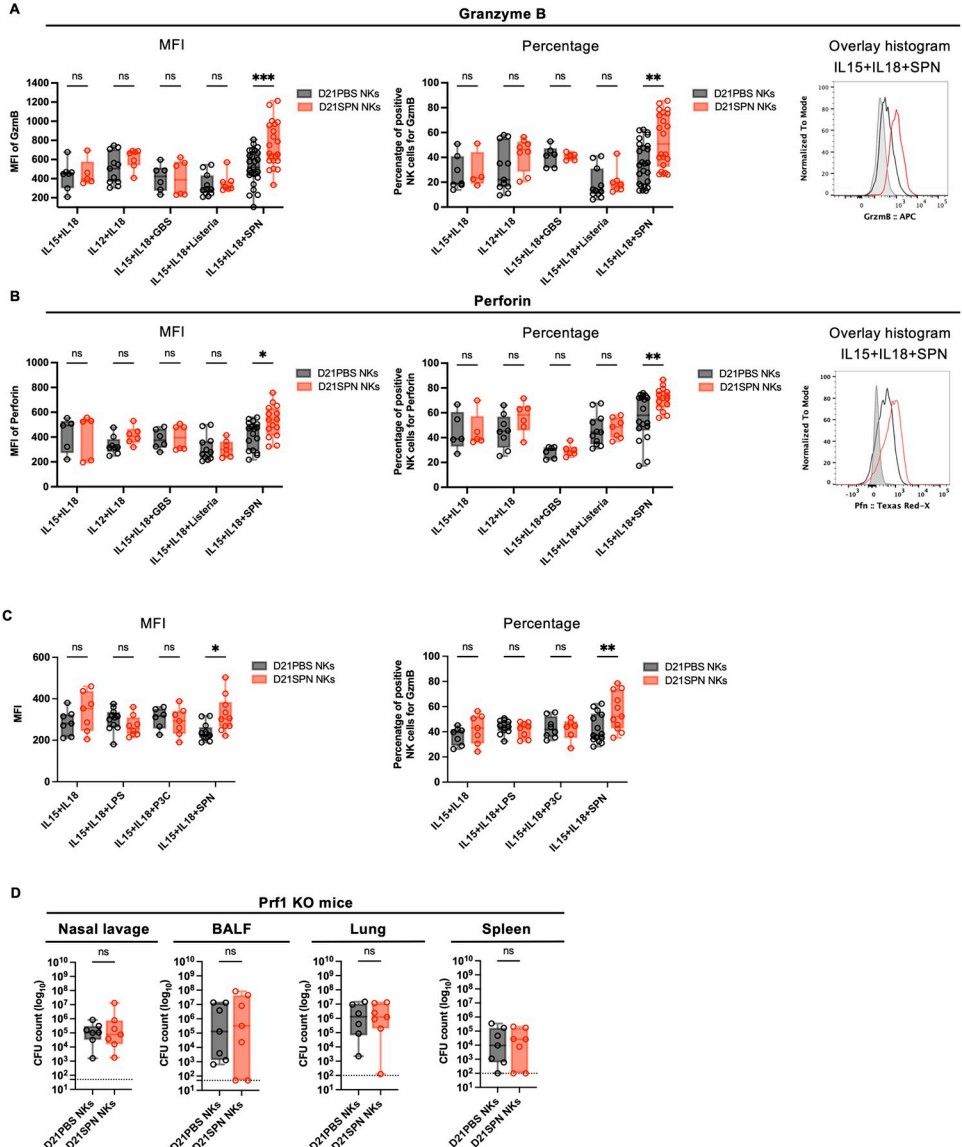

**Fig 6. Cytotoxic proteins are upregulated by memory NK cells and important for protection of mice. (A-B-C)**
C57BL/6 mice were intranasally injected with either PBS (black symbols) or sub-lethal dose of *S. pneumoniae* (SPN,
red symbols, 5x10⁵ CFU) for two consecutive days. After 21 days, NK cells were highly purified from spleens of
D21PBS or D21SPN mice (98% of purity) and stimulated *in vitro* with cytokines (IL-15 at 2 ng/ml, IL-18 at 1,5 ng/ml,
IL-12 at 1,25 ng/ml) and either formaldehyde inactivated bacteria (MOI 20), Lipopolysaccharide (LPS) or the synthetic
lipopeptide Pam3CSK4 (P3C) for 24 hours. SPN: *S. pneumoniae*, GBS: *S. agalactiae*, Listeria: *L. monocytogenes*. **(A)**
Intensity of Granzyme B expression in NK cells (MFI, **left panel**), percentage of Granzyme B⁺ NK cells (**middle
panel**), representative overlay histogram upon IL-15+IL-18+SPN stimulation (**right panel**, gray represents isotype
control). D21PBS NK cells and D21SPN NK cells are purified and pooled from n ≥ 3 mice/group and incubated in
n ≥ 2 experimental replicates/group. Box plots where each dot represents an experimental replicate (black dots for
D21PBS NK cells, red dots for D21SPN NKs cells), lines are median, error bar show min to max. Data are
representative of at least two experiments. **(B)** Intensity of Perforin expression in NK cells (MFI, **left panel**),
percentage of Perforin⁺ NK cells (**middle panel**), representative overlay histogram upon IL-15+IL-18+SPN
stimulation (**right panel**, gray represents isotype control). D21PBS NK cells and D21SPN NK cells are purified and
pooled from n ≥ 3 mice/group and incubated in n ≥ 2 experimental replicates/group. Box plots where each dot
represents an experimental replicate (black dots for D21PBS NK cells, red dots for D21SPN NKs cells), lines are
median, error bar show min to max. Data are representative of at least two experiments. **(C)** NK cells were stimulated
*in vitro* with cytokines, SPN, Lipopolysaccharide (LPS) and the synthetic lipopeptide Pam3CSK4 (P3C). Intensity of
Granzyme B expression in NK cells (MFI, **left panel**), percentage of Granzyme B⁺ NK cells (**right panel**). D21PBS NK
cells and D21SPN NK cells are purified and pooled from n ≥ 4 mice/group and incubated in n ≥ 3 experimental

replicates/group. Box plots where each dot represents an experimental replicate (black dots for D21PBS NK cells, red dots for D21SPN NKs cells), lines are median, error bar show min to max. Data are representative of three experiments. **(D)** Mouse infections are carried out as in the scheme in Fig 3A, with the exception that Prf1 KO mice were used as both donor and recipient mice. Bacterial counts at 40h post-infection in the nasal lavage, bronchio-alveolar lavage fluid (BALF), lungs and spleen of Prf1 KO mice having received D21PBS Prf1 KO NK cells (black symbols) or D21SPN Prf1 KO NK cells (red symbols). Box plots where each dot represents an individual mouse, lines are the mean, error bars show min to max and dotted lines represent limit of detection. Data are pooled from two repeats with n ≥ 3 mice/group. ns, not significant. * $p < 0.05$, ** $p < 0.01$, *** $p < 0.001$. 2way ANOVA **(A,B,C)** and Mann-Whitney **(D)** tests for statistical significance.

NK cell responses are not mediated by TLR receptors sensing of bacteria. These data are in agreement with the specificity of the memory recall responses, and suggest some unknown receptor is involved.

To explore whether heightened cytotoxic function of memory NK cells is associated with epigenetic changes, we assessed the histone mark H3K4me1 associated with upstream regulatory regions of *gzmb* gene in D21PBS and D21SPN NK cells. Although we did not reach statistical significance, we showed a consistent increase of H3K4me1 at -34kb upstream *gzmb* gene in D21SPN NK cells compared to D21PBS NK cells (S5C Fig), which was not observed at the intergenic region (IG). Thus, our results suggest that memory NK cells are associated with a gain of H3K4me1 at specific regulatory loci of *gzmb* gene.

To analyze the role of cytotoxic proteins in the protection of mice, we used Perforin KO mice. First, we generated D21PBS or D21SPN Perforin KO NK cells as previously described in Fig 3A. To control that Perforin KO NK cells are still able to acquire memory properties upon *S. pneumoniae* infection, we stimulated them *in vitro* and showed that D21SPN Perforin KO NK cells produce more Granzyme B than D21PBS Perforin KO NK cells upon stimulation with IL-15+IL-18+SPN (S5D Fig). Next, we transferred highly purified D21PBS or D21SPN Perforin KO NK cells into naïve Perforin KO recipient mice and infected them with a lethal dose of *S. pneumoniae* (scheme in Fig 3A). Coherent with our *in vitro* results, we also observed an increased production of Granzyme B in the lungs of mice having received D21SPN Perforin KO NK cells compared to the control group (S5E Fig). In addition, we evaluated the percentages of neutrophils and NK cells in Perforin KO mice and observed a similar recruitment and activation of cells in both groups of recipient mice (S5F Fig). Importantly, by comparing bacterial counts in the organs of recipient mice we did not observe a reduction of CFU in mice having received D21SPN Perforin KO NK cells compared to control group (Fig 6D). Therefore, our data suggest that memory NK cell protection from lethal infection is abolished in the absence of Perforin production.

## Discussion

In this study, we show that NK cells play an important role in extracellular bacterial infections, a property previously poorly described. Notably, NK cells retain an intrinsic memory of previous bacterial encounters and display heightened responses upon secondary exposure to the same bacterium and protect naïve mice from a lethal infection. Surprisingly, protection does not require IFNγ or recruitment of inflammatory cells, instead, an inherent property of NK cells and production of cytotoxic factors seem to be required.

Although NK cell memory has been described upon various viral infections, such response to bacterial infections remains poorly understood. Upon infection with *Ehrlichia muris*, an intracellular bacterial pathogen, NK cells develop memory-like responses [36]. Indeed, transferred memory-like NK cells from *E. muris*-primed donor mice conferred protection in Rag2$^{-/-}$ IL2rg2$^{-/-}$ recipient mice against a high dose *E. muris* challenge. However, these

memory-like NK cells were not characterized and the mechanisms involved in the protection remain undefined. Also, several studies have investigated NK cell memory upon BCG vaccination but resulted in different conclusions. One study showed that purified splenic NK cells from BCG vaccinated mice do not produce more IFNγ or enhance macrophage phagocytosis in the presence of BCG *in vitro* [37]. However, it has been suggested that a subset of CD27[+] KLRG1[+] NK cells expand following BCG inoculation and confer protection against *M. tuberculosis* infection by reducing the CFU numbers in the lungs at 30 days post-infection [38]. Finally, it has been shown that BCG immunization may induce antigen-independent memory-like NK cells that can protect mice to heterologous challenge with *Candida albicans* [39]. Altogether, these studies, along with our work, indicate that NK cell memory could be a general feature of NK cells to all bacteria, and could have important functions during infection.

It remains undefined how NK cells sense and respond to bacteria. Activating receptors are engaged upon sensing of cells infected with intracellular bacteria [40], however, NK cell interactions with extracellular bacteria are poorly explored. One study on human NK cells reported an engagement between the inhibitory receptor Siglec-7 and the β-protein on the surface of group B Streptococcus [41]. However, neither Siglec-7, nor a homologue, is expressed in mice, and thus NK cell binding-interactions with extracellular bacteria are undefined. In our model, we have shown *in vitro* that purified NK cells can be directly activated upon incubation with only IL-15+IL-18 and *S. pneumoniae*. Therefore, our data strongly suggest that NK cells directly interact with bacteria in a TLR independent manner thereby inducing cell responses. Such interaction could occur through an unknown surface receptor recognizing some component of the pneumococcal capsule or membrane. It should be noted that although NK cells sense bacteria *in vitro* to initiate response, the mechanisms of memory acquisition could be different. We hypothesize that NK cells acquire memory by a direct interaction with pneumococcus, but we cannot rule out the possibility that other immune cells are involved in this process. As in other NK cell memory models [38,42–44], some pro-inflammatory cytokines and costimulatory molecules expressed by other cells might be required for generating *S. pneumoniae* memory NK cells.

Previous studies have shown that memory or memory-like NK cells to viruses and cytokines display increased IFNγ secretion upon re-stimulation [8,45]. Furthermore, endotoxemia-induced memory-like NK cells were also described to produce more IFNγ following *in vivo* secondary LPS injection [35]. Altogether, it suggests that IFNγ production is the main effector function of memory NK cells in these models. However, in our model, although we detected an increase of IFNγ[+] NK cells *in vitro*, the protective effect of memory NK cells was not dependent on IFNγ. In fact, the absence of IFNγ signaling in recipient mice did not affect bacterial dissemination and neutrophils recruitment at early timepoints. Furthermore, we showed less IFNγ[+] NK cells in the lungs of infected mice receiving memory NK cells. Interestingly, NK cells display a slight remodeling of chromatin at an IFNγ enhancer suggesting that at the epigenetic level, memory NK cells are ready to produce more IFNγ upon the proper stimulus, such as that provided *in vitro*. Surprisingly, we find no evidence that IFNγ is playing a role in our mouse model and the reduced production of cytokines in the lungs of protected mice could be the consequence of lower number of bacteria in those animals.

The function of NK cells in the recruitment of inflammatory cells is well defined, so is their role in cytotoxicity of infected cells. Indeed, the release of cytotoxic molecules into the immune synapse to induce apoptosis of infected cells is well described [23,46]. Granules secreted by NK cells contain the pore-forming protein Perforin which damages the target cell membrane allowing Granzymes to pass, resulting in the activation of the caspase pathway and reactive oxygen species (ROS) production. Interestingly, the release of Granulysin and Granzyme B into the cytosol of infected cells targets intracellular bacteria [47]. However, the importance of

NK cell cytotoxicity in the context of extracellular bacterial infections is not well understood. A few intriguing *in vitro* studies report direct receptor-mediated recognition and killing of extracellular bacteria such as *Mycobacterium tuberculosis*, *Burkholderia cenocepacia* or *Pseudomonas aeruginosa* by NK cell degranulation [48–50]. Recognition of bacteria by NK cells has been suggested to be receptor mediated as soluble NKp44 binding to extracellular *Mycobacterium tuberculosis* was shown *in vitro* [51]. Our data suggest a similar finding as TLRs are not activated and responses are specifically directed towards the same bacteria as the primary stimulus. During an *in vivo* infection, neither bacterial recognition nor direct killing has been demonstrated, and the mechanisms at play are unknown. In our model of extracellular bacterial infection *in vivo*, we have shown that memory NK cells lead to more Granzyme B production in the lungs of protected mice, suggesting this effector protein could contribute to host protection against *S. pneumoniae*. Although the pneumococcus lifestyle is primarily extracellular, a few studies have reported occasional intracellular replication within splenic macrophages and lung epithelial cells [52,53]. In our *in vivo* studies, we therefore cannot exclude the possibility that memory NK cells are protecting the host by targeting cells infected with intracellular pneumococci.

NK cell-based therapies are the basis of a new generation of innovative immunotherapy for cancer, which yields encouraging results in clinical trials [54]. It is based on activation of NK cells *ex vivo* for heightened activity against tumor cells. Our work, along with the extensive knowledge on NK cell memory to viruses, could suggest similar *ex vivo* activation strategies against infections. Gaining a greater understanding of the mechanisms at play as well as the memory features will be paramount.

## Materials and methods

### Ethics statement

All protocols for animal experiments were reviewed and approved by the CETEA (Comité d'Ethique pour l'Expérimentation Animale—Ethics Committee for Animal Experimentation) of the Institut Pasteur under approval number dap170005 and were performed in accordance with national laws and institutional guidelines for animal care and use.

### Animal model

Experiments were conducted using C57BL/6J females of 7 to 12 weeks of age. CD45.2 mice were purchased from Janvier Labs (France). CD45.1, *Ifngr1*$^{-/-}$ (JAX stock #003288) and *Prf1*$^{-/-}$ (JAX stock #002407) mice were maintained in house at the Institut Pasteur animal facility.

### Bacterial inocula

All experiments that include infection with *Streptococcus pneumoniae* were conducted with the luminescent serotype 4 TIGR4 strain (Ci49) a commonly used pathogenic serotype (containing pAUL-A Tn*4001 luxABCDE* Km', [55]), obtained from Thomas Kohler, Universität Greifswald. Experimental starters were prepared from frozen master stocks plated on 5% Columbia blood agar plates (Biomerieux ref no. 43041) and grown overnight at 37˚C with 5% $CO_2$ prior to outgrowth in Todd-Hewitt (BD) broth supplemented with 50mM HEPES (TH +H) and kanamycin (50 µg/ml), as previously described in [56]. Inoculum were prepared from frozen experimental starters grown to midlog phase in TH+H broth supplement with kanamycin (50 µg/ml) at 37˚C with 5% $CO_2$ in unclosed falcon tubes. Bacteria were pelleted at 1500xg for 10 min at room temperature, washed three times in DPBS and resuspended in DPBS at the

desired CFU/ml. Bacterial CFU enumeration was determined by serial dilution plating on 5% Columbia blood agar plates.

*Listeria monocytogenes* EGD strain were grown overnight in brain heart infusion (BHI) liquid broth with shaking at 37˚C. Overnight culture were then subcultured 1/10 into fresh BHI and grown to an OD600 of 1. Bacteria were washed three times and subsequently resuspended in DPBS. Experimental starters were then frozen at -80. For mouse infections, experimental starters were thawed on ice and diluted in fresh DPBS at the desired CFU/ml.

For bacteria killed with paraformaldehyde (PFA), the concentrated bacteria, prior to dilution, were incubated in 4% PFA for 30 minutes at room temperature, washed three times in DPBS and diluted to the desired CFU/ml.

*Streptococcus agalactiae* (GBS, NEM 316 strain) were grown in TH+H medium with shaking at 37˚C to an OD600 of 1. Bacteria were pelleted at 1500xg for 10 minutes at room temperature, incubated in 4% PFA for 30 minutes at room temperature, washed three times in DPBS and resuspended in DPBS at the desired CFU/ml.

## Mouse infection

Animals were anaesthetized with a ketamine and xylazine cocktail (intra-muscular injection) prior to infection. Mice were infected with *Streptococcus pneumoniae* by intranasal instillation of 20µl containing $5x10^5$ (sub-lethal dose), $5x10^6$ (survival study) or $1x10^7$ CFU (lethal dose). Control mice received intranasal injection of 20µl of DPBS. For *Listeria monocytogenes* infection, mice were injected with 100µl of $1x10^6$ CFU by retro-orbital injection. Animals were monitored daily for the first 5 days and then weekly for the duration of the experiment. Mouse sickness score from [57] was used to assess progression of mice throughout *S. pneumoniae* infection, representing the following scores: 0- Healthy; 1-transiently reduced response/ slightly ruffled coat/transient ocular discharge/up to 10% weight loss; 2 and 3- (2-up to 2 signs, 3- up to 3 signs) clear piloerection/intermittent hunched posture/persistent oculo-nasal discharge/persistently reduced response/intermittent abnormal breathing/up to 15% weight loss; 4-death. Animals were euthanized by $CO_2$ asphyxiation at $\geq$ 20% loss of initial weight or at persistent clinical score 3.

## Sampling

The nasal lavage was obtained by blocking the oropharynx to avoid leakage into the oral cavity and lower airway, and nares were flushed two times with 500µl DPBS. Bronchio-alveolar fluid (BALF) was collected by inserting catheter (18GA 1.16IN, 1.3x30 mm) into the trachea and washing the lungs with 2ml of DPBS. Blood was collected by cardiac puncture at left ventricle with 20G needle and immediately mixed with 100mM EDTA to prevent coagulation. Spleens were disrupted by mechanical dissociation using curved needles to obtain splenocytes suspension in DPBS supplemented with 0,5% FCS+0,4% EDTA. Lungs were mechanically dissociated with gentleMACS Dissociator (Miltenyi Biotec) in cTubes containing lung dissociation kit reagents (DNAse and collagenase ref no. 130-095-927, Miltenyi Biotec).

Lungs, BALF, nasal lavage, blood, spleen homogenates were plated in serial dilutions on 5% Columbia blood agar plates supplemented with gentamycin (5µg/ml) to determine bacterial counts.

## NK cell purification and transfer

Splenocytes were passed successively through 100µm, 70µm and 30µm strainers (Miltenyi Biotec) in DPBS (+0,5% FCS+0,4% EDTA) and counted for downstream applications. NK cells were first enriched from splenocytes suspensions using negative enrichment kit (Invitrogen,

ref no. 8804–6828) according to manufacturer's protocol but combined with separation over magnetic columns (LS columns, Miltenyi Biotec). NK cells were then re-purified by negative selection using purification kit and magnetic columns according to manufacturer's instructions to reach purities of approximately 98% (Miltenyi Biotec, MS columns and isolation kit ref no. 130-115-818). NK cells were purified from mechanically dissociated lungs (as described above) using the same purification protocol but incubating the cells with anti-CD31, anti-CD326 and antiTer119 microbeads beads (Milteny Biotec refs. 130-097-418, 130-1-5-958 and 130-049-901 respectively) for 5 minutes previous to the second purification protocol. Purified NK cells were transferred intravenously by retro-orbital injections into recipient mice (0,2-1x10^6 cells/mouse) in 100μl of DPBS or cultured for *in vitro* experiments.

## NK cell culture

Purified NK cells were cultured at 1x10^6 cells/ml in 200μl RPMI 1640 (Gibco) supplemented with 10% FCS and 1 U/mL Penicillin-Streptomycin (10 000 U/mL, Gibco) in 96 well round-bottom plates. Cells were either left unstimulated or activated with paraformaldehyde inactivated bacteria at MOI 20 (*S. pneumoniae*, L. *monocytogenes* or *S. agalactiae)* and/or with various cytokine cocktails composed of IL-15 (2 ng/ml), IL-18 (1,5 ng/ml) and IL-12 (1,25 ng/ml) (Miltenyi Biotec). Cells were also stimulated with the TLR1/TLR2 agonist Pam3CSK4 (Invivo-Gen) or with LPS from E. coli O111:B4. After 20 hours of culture at 37˚C, 1X Brefeldin-A (BFA solution 1000X, ref no. 420601, BioLegend) was added into each well to block secretion of IFNγ for 4 hours. After incubation at 37˚C, cells were collected for intracellular cytokine and cytotoxic proteins by flow cytometry.

## Cell preparation for cytometry staining

Following mechanical dissociation, lung and spleen homogenates were passed through 100μm strainers in DPBS supplemented with 0,5% FCS+0,4% EDTA, lysed in 1X red blood cell lysis buffer for 3 minutes (ref no. 420301, BioLegend), and passed successively through 70μm and 30μm strainers (Miltenyi Biotec). Single cell suspensions from BALF, lung and spleen were counted and prepared for surface staining in 96 well plates. Prior to IFNγ staining, single cell suspensions are incubated with 1X Brefeldin-A for 4 hours at 37˚C to block secretion of cytokines (ref no. 420601, BioLegend). For all, cells were first stained with anti-mouse CD16/CD32 to block unspecific binding and a cocktail of surface labeling antibodies for 40 minutes in DPBS (+0,5% FCS+0,4% EDTA). Next, cells were stained for viability using fixable viability dye (eFluor780, ref no. 65-0865-14, Invitrogen) for 5 minutes at 4˚C and then fixed using commercial fixation buffer for 3 minutes (ref no. 420801, BioLegend). For intracellular staining of Granzyme B and Perforin, cells were permeabilized with a Fixation/Permeabilization commercial kit for 30 minutes (Concentrate and Diluent, ref no. 00-5123-43, Invitrogen). After permeabilization and wash, cells were stained with respective antibodies in DPBS (+0,5% FCS +0,4% EDTA) overnight at 4˚C. For intracellular staining of IFNγ, cells were permeabilized and stained in buffer from commercial kit (Inside Stain Kit, ref no. 130-090-477, Miltenyi Biotec) for 40 minutes at 4˚C. After a final wash suspended in DPBS, sample acquisitions were performed on MACSQuant (Miltenyi Biotec) and LSRFortessa (BD Biosciences) flow-cytometers and analysis were done using FlowJo Software (TreeStar).

## ELISA

Following mechanical dissociation, lung homogenates are centrifuged at 400xg for 7 minutes at 4˚C. Supernatants are collected and frozen at -20˚C for cytokine assays performed by ELISA according to manufacturer's instructions (DuoSet, R&D Systems).

## ChIP-qPCR assays

Around $4 \times 10^6$-$6 \times 10^6$ NK cells were fixed in 1% formaldehyde (8 min, room temperature), and the reaction was stopped by the addition of glycine at the final concentration of 0,125 M. After two washes in PBS, cells were resuspended in 0.25% Triton X-100, 10 mM Tris-HCl (pH 8), 10 mM EDTA, 0.5 mM EGTA and proteases inhibitors; the soluble fraction was eliminated by centrifugation; and chromatin was extracted with 250 mM NaCl, 50 mM Tris-HCl (pH 8), 1 mM EDTA, 0.5 mM EGTA and proteases inhibitors cocktail for 30 min on ice. Chromatin was resuspended in 1% SDS, 10 mM Tris-HCl (pH 8), 1 mM EDTA, 0.5 mM EGTA and proteases inhibitors cocktail; and sonicated during 10 cycles using Diagenode Bioruptor Pico (30 sec on 30 sec off). DNA fragment size (<1 kb) was verified by agarose gel electrophoresis. ChIP was performed using H3K4me1 antibody and nonimmune IgG (negative control antibody), 10 μg chromatin per condition was used. Chromatin was diluted 10 times in 0.6% Triton X-100, 0.06% sodium deoxycholate (NaDOC), 150 mM NaCl, 12 mM Tris-HCl, 1 mM EDTA, 0.5 mM EGTA and proteases inhibitors cocktail. For 6 hours, the different antibodies were previously incubated at 4˚C with protein G-coated magnetic beads (DiaMag, Diagenode), protease inhibitor cocktail and 0.1% BSA. Chromatin was incubated overnight at 4˚C with each antibody/protein G-coated magnetic beads. Immunocomplexes were washed with 1 x buffer 1 (1% Triton X-100, 0.1% NaDOC, 150 mM NaCl, 10 mM Tris-HCl (pH 8)), 1 x buffer 2 (0.5% NP-40, 0.5% Triton X-100, 0.5 NaDOC, 150 mM NaCl, 10 mM Tris-HCl (pH 8)), 1 x buffer 3 (0.7% Triton X-100, 0.1% NaDOC, 250 mM NaCl, 10 mM Tris-HCl (pH 8)) 1 x buffer 4 (0.5% NP-40, 0.5% NaDOC, 250 mM LiCl, 20 mM Tris-HCl (pH 8), 1 mM EDTA) and 1 x buffer 5 (0,1% NP-40, 150 mM NaCl, 20 mM Tris-HCl (pH 8), 1 mM EDTA). Beads were eluted in water containing 10% Chelex and reverse cross-linked by boiling for 10 min, incubating with RNase for 10 min at room temperature, then with proteinase K for 20 min at 55˚C and reboiling for 10 min. DNA fragment were purified by Phenol Chloroform extraction. Amplifications (40 cycles) were performed using quantitative real-time PCR using Universal Syber Green Supermix (BIORAD) on a CFX384 Touch Real-Time PCR system (BIORAD). qPCR efficiency (E) was determined for the ChIP primers with a dilution series of genomic mouse DNA. The threshold cycles (Ct values) were recorded from the exponential phase of the qPCR for IP and input DNA for each primer pair. The relative amount of immunoprecipitated DNA was compared to input DNA for the control regions (% of recovery) using the following formula:

% recovery = $E\verb|^|((Ct(1\% \text{ input}) - Log_2(\text{input dilution})) - Ct(IP)) \times 100\%$

**Primer sequences.** -22kb enhancer of the IFNγ locus
5'CCAGGACAGAGGTGTTAAGCCA3'
5'GCAACTTCTTTCTTCTCAGGGTG3'
-55kb enhancer of the IFNγ locus
5'GGCTTCCTGTCATTGTTTCCA3'
5'CAGAGCCATGGGATGACTGA3'
-34kb enhancer of Granzyme B locus
5'CCCTCCCCTCTAATCACACA3'
5'ACGGTGTTGAGGGAGTTTCA3'
Intergenic Region (IG)
5'CCACACCTCTTCCTTCTGGA3'
5'ATTTGTGTCAGAGCCCAAGC3'

**Quantification and statistical analysis.** Statistical significance was tested using Prism 9 Software (GraphPad). Mann Whitney test was used for single comparisons, Kruskal-Wallis test for 3 groups- comparisons, 2way ANOVA for multiple comparisons and Log-rank (Mantel-Cox) test for survival curves.

## Supporting information

**S1 Fig. Receptor expression profile of NK cells following S. pneumoniae infection. (A)** Representative contour plots for NK cell purity of D21PBS (black symbols) and D21SPN (red symbols) samples based on NK1.1$^+$ CD3$^-$ or NK1.1$^+$ DX5$^+$ markers (**left panel**). Percentage of NK cells among live cells (purity of cells, **right panel**). Box plots where each dot represents a pool of mice from one experiment, lines are the median, error bars show min to max. Data are representative of more than three repeats with n ≥4 pooled mice/group and n ≥ 3 experimental replicates/group. **(B)** Mouse infections are carried out as in the scheme in Fig 1A. NK cells were isolated from spleens of mice previously infected with *S. pneumoniae* (red bars, D21SPN NKs) or not (black bars, D21PBS NKs). Highly purified NK cells were fixed, and chromatin was extracted and sheared. ChIP for H3K4me1 were performed and resulting positive fractions of the chromatin were amplified using PCR for the indicated targets. Enrichment percentage for H3K4me1 pull-down on regions upstream the *ifng* gene in NK cells from D21PBS and D21SPN mice. Data are representative of four experiments with n ≥ 4 mice/group. **(C-D)** Splenocytes were harvested from mice 12 days **(C)** or 21 days **(D)** after they were intranasally injected with either PBS (black symbols) or sub-lethal dose of *S. pneumoniae* (SPN, red symbols, 5x10$^5$ CFU) for two consecutive days. NK cell expression of several NK cell receptors in percentages (**left panel**) and intensity of expression (MFI, **right panel**). Box plots where each dot represents an individual mouse, lines are the median, error bars show min to max. Data are pooled from one **(C)** or two **(D)** repeats with n ≥ 4 mice/group. ns, not significant. $^*$ p < 0.05 and $^{**}$ p <0.01. Mann-Whitney **(A,B)** and 2way ANOVA **(C,D)** tests for statistical significance.
(TIFF)

**S2 Fig. Extended characterization of NK cell response to primary S. pneumoniae infection.** Weight **(A)** and clinical score **(B)** following the infection scheme from Fig 2A. Dots represent the mean and error bars are the standard error of the mean (SEM). Data are pooled from two repeats with n = 4 mice/group. **(C-E)** Organs were collected at 24h **(C)**, 72h **(D)** and 21 days post-infection **(E)**. Absolute numbers of neutrophils (CD11b$^+$ Ly6G$^+$) in the lungs and bronchio-alveolar lavage fluid (BALF) (**left panel**). Absolute numbers of NK cells (NK1.1$^+$ CD3$^-$), percentage of CD69$^+$ NK cells, intensity of Granzyme B and Perforin expression (MFI) and representative overlay histogram of Granzyme B and Perforin staining in NK cells (**right panel**). Box plots where each dot represents an individual mouse (black dots for uninfected mice, red dots for infected mice), lines are the median, error bar show min to max. Grey histogram represents isotype control. Data are pooled from two or three repeats with n ≥ 3 mice/group. ns, not significant. $^*$ p < 0.05, $^{**}$ p < 0.01. Mann-Whitney test for statistical significance.
(TIFF)

**S3 Fig. Measure of transferred NK cell circulation. (A-B)** Mouse infections are carried out as in the scheme in Fig 3A. Congenic mice are used to distinguish donor NK cells (CD45.2$^+$) from recipient NK cells (CD45.1$^+$). **(A)** Representative gating strategy to measure purity of transferred D21PBS NKs (black symbols) or D21SPN NKs (red symbols) in the lungs of recipient mice at 24 hours post-infection. Purity represents the percentage of NK cells among CD45.2$^+$ transferred cells. Box plots where each dot represents an individual recipient mouse, lines are the median, error bar show min to max. Data are representative of three repeats with n ≥ 4 mice/group. **(B)** Representative gating strategy to detect transferred NK cells. Percentages and numbers of CD45.2$^+$ transferred NK cells in the lungs and blood of CD45.1$^+$ recipient mice at 24h and 40h post-infection. Box plots where each dot represents an individual

recipient mouse, lines are the median, error bar show min to max. Data are pooled from at least two repeats with n ≥ 3 mice/group. ns, not significant. Mann-Whitney test for statistical significance.
(TIFF)

**S4 Fig. Impact of transferred memory NK cells in innate immune cell activation upon S. pneumoniae infection.** Mouse infections are carried out as in the scheme in Fig 3A. Organs were collected at 24h and 40h post-infection for flow cytometry analysis and ELISA assays. **(A)** Gating strategy used to identify myeloid-cell subsets in the lungs at 24h post-infection, adapted from [58]. After the exclusion of debris, doublets and dead cells, immune cells were identified by CD45 staining. Neutrophils, alveolar macrophages and eosinophils are defined with the following specific markers respectively: Ly6G$^+$ CD11b$^+$, SiglecF$^+$ CD11b$^-$, SiglecF$^+$ CD11b$^+$. Gating on CD11b$^{high}$ was used to distinguish myeloid cells from lymphoid cells with the exception of CD103$^+$ dendritic cells that are defined as CD11b$^{low}$ CD103$^+$ CD24$^+$ cells. In CD11b$^{high}$ subset, gating on MHCII$^+$ cells allow to identify interstitial macrophages as MHCII$^+$ CD64$^+$ CD24$^-$ cells and CD11b$^+$ dendritic cells as MHCII$^+$ CD64$^-$ CD24$^+$ cells. On the contrary, CD11b$^{high}$ MHCII$^-$ cells are monocytes/immature macrophages that can have different maturation states based on Ly6C marker. **(B)** Cellular activation analysis in the lungs at 40h post-infection. Intensity of CD11b expression in neutrophils (MFI), percentage of ROS$^+$ neutrophils and intensity of MHCII expression in interstitial macrophages (MFI). Box plots with each dot representing individual mice (blue dots for uninfected mice, black dots for mice having received D21PBS NKs, red dots for mice having received D21SPN NKs), lines are the median, error bar show min to max. Data are pooled from two repeats with n ≥ 1 mice/group. **(C)** ELISA assays of lung supernatants from infected mice having received either D21PBS NKs (black) or D21SPN NKs (red) at 24h post-infection. Bars are the mean of at least 3 experiments with n ≥ 3 mice/group, error bars are the standard error of the mean (SEM). **(D)** Granzyme B ELISA assays of lung supernatants from infected mice having received either D21PBS NKs (black) or D21SPN NKs (red) at 24h post-infection. Bars are the mean of at least 3 experiments with n ≥ 3 mice/group, error bars are the standard error of the mean (SEM). ns, not significant. Kruskal-Wallis **(B),** 2way ANOVA **(C)** and Mann-Whitney **(D)** tests for statistical significance.
(TIFF)

**S5 Fig. Granzyme B expression in memory NK cells.** (A) NK cells were stimulated *in vitro* with cytokines and formaldehyde inactivated SPN (MOI 20) for 24 hours. Percentage and intensity (MFI) of CD11b in NK cells (left panel), percentage and intensity (MFI) of CD27 in NK cells (right panel) in both Granzyme B$^+$ and Granzyme B$^-$ cells. D21PBS NK cells and D21SPN NK cells are purified and pooled from n ≥ 4 mice/group and incubated in n ≥ 3 experimental replicates/group. Box plots where each dot represents an experimental replicate (black dots for D21PBS NK cells, red dots for D21SPN NKs cells), lines are median, error bar show min to max. Data are representative of three experiments. (B-C) C57BL/6 mice were intranasally injected with either PBS (black symbols) or sub-lethal dose of *S. pneumoniae* (SPN, red symbols, 5x10$^5$ CFU) for two consecutive days. After 21 days, NK cells were highly purified from spleens of D21PBS or D21SPN mice (98% of purity) and stimulated *in vitro* with cytokines (IL-15 at 2 ng/ml, IL-18 at 1,5 ng/ml, IL-12 at 1,25 ng/ml) and either formaldehyde inactivated SPN (MOI 20), Lipopolysaccharide (LPS) or the synthetic lipopeptide Pam3CSK4 (P3C) for 24 hours. (B) NK cells were stimulated *in vitro* with cytokines, Lipopolysaccharide (LPS) and the synthetic lipopeptide Pam3CSK4 (P3C) for 24h. Percentage of IFNγ$^+$ NK cells. D21PBS NK cells and D21SPN NK cells are purified and pooled from n ≥ 3 mice/group and incubated in n ≥ 2 experimental replicates/group. Box plots where each dot represents an

experimental replicate (black dots for D21PBS NK cells, red dots for D21SPN NKs cells), lines are median, error bar show min to max. Data are representative of two experiments. (C) Mouse infections are carried out as in the scheme in Fig 1A. NK cells were isolated from spleens of mice previously infected with *S. pneumoniae* (red bars, D21SPN NKs) or not (black bars, D21PBS NKs). Highly purified NK cells were fixed, and chromatin was extracted and sheared. ChIP for H3K4me1 were performed and resulting positive fractions of the chromatin were amplified using PCR for the indicated targets. Enrichment percentage for H3K4me1 pull-down on -34kb upstream the *gzmb* gene and intergenic regions (IG) in NK cells from D21PBS and D21SPN mice. Data are representative of at least three experiments with n ≥ 4 mice/group. (D) Mouse infections and *in vitro* experiments are carried out as in the scheme in Fig 1A, with the exception that Prf1 KO mice were used. Purified D21PBS (black) or D21SPN (red) Prf1 KO NK cells were stimulated *in vitro* with cytokines (IL-15 at 2 ng/ml and IL-18 at 1,5 ng/ml) and formaldehyde inactivated *S. pneumoniae* (SPN, MOI 20) for 24 hours. Intensity of Granzyme B expression in NK cells (MFI, left panel) and percentage of Granzyme B$^+$ NK cells (right panel) upon *in vitro* stimulation. Box plots where each dot represents a pool of mice from one experiment, lines are median, error bar show min to max. Data are representative of one experiment with n = 4 pooled mice/group and n ≥ 2 experimental replicates/group. (E-F) Mouse infections and *in vivo* transfers are carried out as in the scheme in Fig 3A, with the exception that Prf1 KO mice were used as both donor and recipient mice. (E) Granzyme B ELISA experiment of lung supernatants from infected mice at 40h post-infection, having received either D21PBS (black) or D21SPN (red) Prf1 KO NK cells. Dots represent individual mouse, bars are the mean and error bars are the standard error of the mean (SEM). Data are representative of one experiment with n ≥ 3 mice/group. (F) Percentages of neutrophils, CD69$^+$ neutrophils and NK cells in the lungs of Prf1 KO recipient mice having received either D21PBS (black) or D21SPN (red) Prf1 KO NK cells. Box plots with each dot representing individual mouse, line are the median, error bar show min to max. Data are representative of one experiment with n ≥ 3 mice/group. ns, not significant. *** p < 0.001, *** p < 0.0001. Mann-Whitney (A,D,E,F) and 2way ANOVA (B-C) tests for statistical significance.
(TIFF)

## Acknowledgments

We thank P. Bousso (Institut Pasteur) and M. Lecuit (Institut Pasteur) for their generous gifts of *Ifngr*1$^{-/-}$ and *Prf*1$^{-/-}$ mice. We thank C. Werts for her advice and for providing LPS and S. Dramsi for providing GBS. We are thankful to M. Ingersoll for her precious advice throughout the project and corrections of the manuscript. We would like to thank E. Gomez Perdiguero for her help with flow cytometry analysis. We thank P.H. Commere and S. Megharba (Cytometry and Biomarkers UTechS, Institut Pasteur) for their help with flow cytometry acquisition.

## Author Contributions

**Conceptualization:** Tiphaine M. N. Camarasa, Júlia Torné, Christine Chevalier, Orhan Rasid, Melanie A. Hamon.

**Data curation:** Tiphaine M. N. Camarasa, Júlia Torné, Christine Chevalier.

**Formal analysis:** Tiphaine M. N. Camarasa, Júlia Torné, Orhan Rasid.

**Funding acquisition:** Tiphaine M. N. Camarasa, Júlia Torné, Melanie A. Hamon.

**Investigation:** Tiphaine M. N. Camarasa, Júlia Torné.

**Methodology:** Tiphaine M. N. Camarasa, Christine Chevalier.

**Project administration:** Melanie A. Hamon.

**Resources:** Melanie A. Hamon.

**Supervision:** Orhan Rasid, Melanie A. Hamon.

**Writing – original draft:** Tiphaine M. N. Camarasa, Orhan Rasid, Melanie A. Hamon.

**Writing – review & editing:** Tiphaine M. N. Camarasa, Júlia Torné, Melanie A. Hamon.

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
