## [Decision Letter · Decision Letter 0]

28 Feb 2023

Dear Dr. Hamon,

Thank you very much for submitting your manuscript "Streptococcus pneumoniae drives specific and lasting Natural Killer cell memory" for consideration at PLOS Pathogens. As with all papers reviewed by the journal, your manuscript was reviewed by members of the editorial board and by several independent reviewers. In light of the reviews (below this email), we would like to invite the resubmission of a significantly-revised version that takes into account the reviewers' comments. Specifically there was concern over the robustness of some of the conclusions being drawn given the relatively low number of replicates in some of the in vivo studies. In addition it is suggested that more careful consideration be given to the phenotype of the particular NK subset that may be exerting these protective memory effects.

We cannot make any decision about publication until we have seen the revised manuscript and your response to the reviewers' comments. Your revised manuscript is also likely to be sent to reviewers for further evaluation.

Sincerely,

Rachel M McLoughlin, PhD

Academic Editor

PLOS Pathogens

Marcel Behr

Section Editor

PLOS Pathogens

Kasturi Haldar

Editor-in-Chief

PLOS Pathogens

orcid.org/0000-0001-5065-158X

Michael Malim

Editor-in-Chief

PLOS Pathogens

orcid.org/0000-0002-7699-2064

Reviewer's Responses to Questions

**Part I - Summary**

Reviewer #1: This study presents new evidence for protective NK cell memory in the context of respiratory tract infection with S. pneumoniae. Their findings suggest that memory NK cells from the spleen improve protection against lethal S. pneumoniae infection, and this correlates with increased granzyme B production by lung NK cells. Despite the elevation in IFN-gamma production by spleen NK memory cells in response to S. pneumoniae exposure in vitro, IFN-gamma signaling was not required for NK cell memory protection, confirmed using IFNGR-/- mice. Instead, NK cell memory protection was lost in Prf1-/- mice, suggesting importance for NK cell cytotoxic capacity in memory-induced production. This unexpected finding indicates that a unique mechanism related to the expression of perforin and granzyme B is involved in protective NK cell memory against S. pneumoniae infection, in contrast to other examples of NK cell memory which are associated with NK cell IFN-gamma production. The requirement for lung vs spleen NK cell intrinsic changes could be further clarified, as all transfer experiments demonstrating memory NK cell protection use spleen NK cells rather than lung (see additional comments). The study falls short of demonstrating the mechanism by which memory NK cell cytotoxicity, or other cells affected by this response, lead to improved bacterial clearance. Regardless, these findings demonstrate a novel innate immune memory pathway of protection against S. pneumoniae. The data are clearly presented with appropriate controls, the authors have acknowledged relevant literature relating to this work, and conclusions are not over-stated.

Reviewer #2: Camarasa et al describe a form of innate NK cell memory that develops in mice following sub-lethal infection with the bacterial pathogen Streptococcus pneumoniae. The novelty of the study is in the clear demonstration of a memory phenotype in NK cells, subsequent to infection with a predominantly extracellular pathogen. This is interesting of itself, but the finding that the mechanism of NK cell-mediated protection against severe infection is perforin-dependent is perhaps the most eye catching part of the study and challenges a prevailing dogma. Although the authors do not define the mechanism(s) by which perforin production affords protection, they do show convincingly that NK cell production of cytotoxins contributes to immune defence against pneumococci.

The manuscript is well written throughout, follows a logical progression, and is largely experimentally sound. I have a few reservations about certain experiments and some of the discussion lacks depth, but I think the study is robust, has merit and will be of interest to a broad readership.

Reviewer #3: In their manuscript ”Steptococcus pneumoniae drives specific and lasting Natural Killer Cell memory, Camarasa et al. used a mouse model to identify NK cell memory following sublethal infection with extracellular bacteria. In detail, the authors demonstrate long-lasting functional priming of splenic NK cells that is dependent on perforin but not IFN-g. The authors also show that the NK cell recall response is specific to SPN and does not extend to other intracellular bacterial infections such as listeria monocytogenes.

This is a very interesting and relevant study as it sheds novel light on the definition of memory NK cell subsets. In general, the paper is clear and concise, however more interesting results may be observed by extrapolating the data to distinct NK subsets, rather than bulk NK (based on CD49b and CD27 expression). This may lead to the identification of a more specific NK cell subset within which the memory resides. However, some statements are over interpreted and should be reviewed and modified.

**Part II – Major Issues: Key Experiments Required for Acceptance**

Reviewer #1: 1. It is unclear whether lung NK cells from 21d post-infection respond similarly to in vitro stimulation as the spleen NK cells. For example, do they produce IFN-gamma and granzyme B upon stimulation with IL-15/IL-18/SPN?

2. Are there any transient NK cell responses (including IFN-gamma, granzyme B) in spleen NK cells at 24-72 hpi like there are in the lung (in Fig 2B-C)? This would relate to the infection-associated phenotype in spleen NK cells, which are used for the adoptive transfers demonstrating memory NK cell protection.

Reviewer #2: The study contains plenty of high quality data, derived from well-powered experimental analyses. However, a couple of important conclusions are reached based on data trends that don't reach statistical significance. The two sections where histone modification were assessed both suggest interesting (and potentially functionally important) changes that may be driven by infection. However, in both cases, the sample size for analysis is small (n=4 and n=5), which prevents firm conclusions from being drawn. I would suggest the authors attempt an extra few replicates, to pin down whether the mechanisms they describe are driven by epigenetic modifications in ifng and gzmb. Similarly, the differences in mouse survival described in Figure 3C are based on experiments with n=4 per group. This is a very small number for a survival experiment and I would have more confidence in the conclusions reached, if the results were reproduced in a larger sample size.

The adoptive transfer experiments performed with the ifngr KO mice are nicely conceived, demonstrating that IFNg is not the basis of the protective mechanism at play in NK cell memory of pneumococcal infection. I found the experiments with the perforin KO less convincing. Why were the prf1 KO mice used as recipients in these experiments? Transfer of prf1 KO NK cells with a memory phenotype into WT mice might have been used to demonstrate that perforin production by memory cells was the basis of protection. Transfer into the prf1 KO leaves open the possibility that perforin production by non-memory cells might contribute to protection against pneumococcal infection. What was the trajectory/outcome of infection in the prf1 KO animals? Do they experience worse outcomes or harbour higher bacterial burdens than WT?

Reviewer #3: 1. The infection with SPN is intranasal, and the target organ for SPN is the lung. However, the authors used splenic NK cells for adoptive transfer experiments. Could similar results be obtained with NK cells from other organs, in particular lung?

2. Fig. 1: Do NK cells also respond to SPN + IL-15 alone (no IL-18)? How important is an inflammatory environment, mimicked by IL-18, for the NK cell response in presence of SPN?

3. Fig. 1: Do NK cells respond with a similar response if treated with another formaldehyde-inactivated bacterium?

4. Fig. 1: How do NK cells respond to live, noninactivated SPN?

5. SPN-mediated activation of myeloid cells has been shown to be TLR2- and TLR4-mediated. The authors should identify whether this is also true for memory NK cells. Is the recall response by NK cells altered if the respective receptors are blocked or knocked out?

6. The authors suggest that the memory NK cell response is specific to the first pathogen the mice have been infected with (SPN). It would be highly relevant to identify the reason for this specificity. Do NK cells depend the on the same PRRs for responding to SPN and to L. monocytogenes? Furthermore, the main target organs for SPN and L. monocytogenes differ (lung vs liver). Can the same ‘specific’ NK cell memory response as e.g. in Fig. 1 be confirmed if mice were infected with L. monocytogenes instead of SPN and NK cells re-stimulated with the same pathogen? This would reveal important information and confirm the authors’ statements concerning ‘specificity’ of the memory NK response.

7. The authors should extend the phenotypic characterization of the responding NK cells (vs non-responding NK cells), e.g. NK cell differentiation/maturation, Ki67. Since only bulk NK cells are compared, it may be that the differences are hidden when looking at this level which may be revealed when gating further down on NK cell subsets based on expression of Ly6C+, CD27low/neg, CD11b+ for example (Sun et al., Nature. 2009., Schuster et al., Immunity. 2023)

8. Fig. S1C: A significant difference for the MFI of Ly49D is not visually clear from the data and difficult to believe – the authors should both increase the number of experiments and provide representative plots/histograms in order to confirm their statement.

9. Fig. S2: The percentage of CD69+ NK cells in the lung seems overall rather low, and after 72h, it decreases even further. The authors should provide representative stainings for CD69 as well as data for CD69 expression before infection – does the frequency increase after 24h compared to before (mock-)infection, or is CD69 expression rather decreased for some reason at 72h and 21d? Furthermore, since higher percentages of CD69+ NK cells at 24h are even present in the PBS-control mice, it seems unlikely that this is an effect due to the infection. These data are confusing, and the authors must be careful with their conclusion of a low-level immune response. In relation to this, the authors should also reveal why granzyme B seems to increase in the control group (see also comment below).

10. Fig. 2 and S2: From Fig. S2, no clear granzyme B signal is detectable in comparison to the isotype control (the few ‘positive’ events are rather likely an effect by spillover from other channels since I assume that this was not a FMO ctrl?). While it might be possible that lung NK cells express less granzyme B, splenic NK cells were found to also express granzyme B according to Fig. 2D, which is not supported by the representative data in Fig. S2D. In particular at 21d, Fig. 2D shows a clear percentage of granzyme B-positive cells in lung and spleen, which is not confirmed by the representative overlays in Fig. S2D. This is confusing, and the authors need to present more reliable data to support their statements concerning granzyme B expression. In relation to this, the authors also need to present representative data for perforin expression at the different timepoints and groups in order to support their statements.

11. It is unclear why only splenic NK cells have been analyzed in their phenotype (Fig. S2). The authors should add analyses on NK cells from other organs, in particular the lungs.

12. Fig. 4A: It would be interesting to see whether the percentage of IFN-g+ NK cells further increases e.g. at 72h, or whether the peak of the response of D21SPN NK cells is reached earlier.

13. The number of experiments should be increased for several datasets throughout the manuscript. In general, more than just one experiment should be performed for each figure. It is also not entirely clear why the authors sometimes show the data as pooled experiments, and in other plots each individual mouse. This is confusing, and the paper would benefit from a more consistent data presentation.

14. The authors state that ‘transferred congenic CD45.2+ NK cells were circulating and detectable at similar percentages in lungs and blood, suggesting there is no preferential trafficking between D21PBS and D21SPN NK cells’. As the lungs are perfused extensively with blood, any differences in NK cell number/percentages could be completely diluted and therefore missed. The lungs could be flushed to clear the blood and then stained to determine NK cell numbers/ percentages. Alternatively, fluorescently labelled anti-CD45 could be given IV before analysis to show what is actually circulating and what may be resident in the parenchyma instead. Again, subset specification may also pull out more interesting results

15. Fig. 5D: The reduction of CXCL1 in the D21SPN NK cells indicates that this chemokine was e.g. consumed by cells infiltrating the lung, hence, this is not a clear indicator for the lack of NK cell infiltration. The authors should combine this analysis with the expression patterns of the respective chemokine receptors on NK cells.

16. Fig. 6: The authors here show a population of granzyme B+ as well as perforin+ NK cells. Does the phenotype differ to the respective negative NK population?

**Part III – Minor Issues: Editorial and Data Presentation Modifications**

Reviewer #1: 1. Fig 2B appears to be from x1 experiment. Are these data representative from at least 2 repeats? If not, they should be repeated.

2. In figure legends (ex, Fig 1) it is unclear what "each dot represents a pool of mice" refers to. Is this meant to be pool of cells from one mouse?

3. Why is there high baseline granzyme B detected for 21 day NK cells (Fig 2D) vs other panels?

4. Fig 3B update legend to include 24 h.

5. Discussion pg 9, Fig 4C results update to reflect that there are reduced burdens in IFNGR-/- mice (line 214), don't see significantly reduced lung bacteria (line 217).

6. Methods are missing for GBS growth.

Reviewer #2: Pneumococcus is referred to throughout the manuscript as an extracellular pathogen. This is surely its primary lifestyle, but many studies have shown the ability of pneumococci to access intracellular compartments and demonstrated that intracellularity, whilst rare, can make important contributions to infection outcomes (see https://pubmed.ncbi.nlm.nih.gov/29662129/, https://pubmed.ncbi.nlm.nih.gov/33216805/). Some discussion could be added, regarding the possibility that NK cell memory might be mediating protection via targeting intracellular subpopulations of bacteria.

The authors describe bacterial sensing by NK cells as the mechanism behind the memory phenotype they observe. Indeed, the data they present supports some contribution from direct sensing mechanisms, but the possibility remains that the original cue for memory NK cell responses to develop in vivo requires other immune cells. Previous studies have suggested that inflammasome-driven macrophage responses can promote NK cell memory (https://pubmed.ncbi.nlm.nih.gov/27287410/), whilst others have shown the pneumococcal infection drives inflammasome responses that ultimately lead to an NK cell IFNg response in the lung (https://pubmed.ncbi.nlm.nih.gov/21085613/). Some discussion of these points in the context of the authors' own findings would be welcome.

A little extra clarity on the murine models is needed in places. What is the rationale behind the consecutive dosing, over two days, when performing infections? Does NK cell memory require this double dose? In Figure 3, was the dose 1x10^7 (line 167) or 5x10^6 (line 174)?

Line 112: Replace 'than' with 'as'.

Reviewer #3: 1. The response of NK cells is dependent on the SPN-serotype. Why did the authors select serotype 4 for their study, and is NK cell memory function detectable with other SPN serotypes? The authors should discuss this.

2. The authors should mention that the results are derived from a mouse model in the abstract.

3. When the authors write e.g. 2.105, I assume they mean 2x105? Please revise throughout the manuscript.

4. In line 158 the authors state that after 21 days post infection that immune cells are similar to uninfected controls ‘both in their number and activity’. No activity was measured. The authors should modify this sentence.

5. The authors repeatedly state that they isolate or stimulate ‘highly purified memory NK cells’, which is not the case. They are purifying bulk NK cells which probably contain memory NK cells to differing degrees. Therefore, although the NK cell population may be highly pure (they state >98% purity), this does not mean they are purifying memory cells. Purifying memory cells suggests they are identifying a distinct NK population and they are not. The authors should revise these statements.

6. In Fig. 5C and S4B, only a handful of innate cells are described. This is not exhaustive and does not include any adaptive cells, therefore their conclusion stating ‘protection provided by memory NK cells is not through enhanced recruitment or activation of inflammatory cells’ is over-interpreted. This sentence should be modified. For example, NK cells are known to recruit CD8+ T cells which were not quantified, neither were inflammatory monocytes which are common inflammatory cells recruited upon infection.

PLOS authors have the option to publish the peer review history of their article (what does this mean?). If published, this will include your full peer review and any attached files.

Reviewer #1: **Yes: **Sarah E Clark

Reviewer #2: No

Reviewer #3: No
---

## [Decision Letter · Decision Letter 1]

27 Jun 2023

Dear Dr. Hamon,

We are pleased to inform you that your manuscript 'Streptococcus pneumoniae drives specific and lasting Natural Killer cell memory' has been provisionally accepted for publication in PLOS Pathogens.

Before your manuscript can be formally accepted you will need to complete some minor corrections as requested by the reviewers below and also complete some formatting changes, which you will receive in a follow up email. A member of our team will be in touch with a set of formatting requests.

Best regards,

Rachel M McLoughlin, PhD

Academic Editor

PLOS Pathogens

Marcel Behr

Section Editor

PLOS Pathogens

Kasturi Haldar

Editor-in-Chief

PLOS Pathogens

orcid.org/0000-0001-5065-158X

Michael Malim

Editor-in-Chief

PLOS Pathogens

orcid.org/0000-0002-7699-2064

Reviewer Comments (if any, and for reference):

Reviewer's Responses to Questions

**Part I - Summary**

Reviewer #1: The revised manuscript addresses all major concerns regarding expanding the comparison of lung versus splenic NK cells for the model system of protective memory NK cells and has increased rigor by including additional experimental replicates for several figures.

Reviewer #2: The authors have addressed my comments in full. Not all suggested experiments have been performed, and I am still not entirely convinced by the ChIP-qPCR data, but I accept the authors' arguments that the experiments are not ethically justifiable, given the large number of animals needed. I agree, also, that the weak effect is likely because bulk NK cells were used for the assays, rather than the memory cell subset.

Reviewer #3: All questions have been answered by the authors in a satisfactory way.

**Part II – Major Issues: Key Experiments Required for Acceptance**

Reviewer #1: (No Response)

Reviewer #2: (No Response)

Reviewer #3: (No Response)

**Part III – Minor Issues: Editorial and Data Presentation Modifications**

Reviewer #1: Please define MFI (mean vs median) used.

Reviewer #2: The response of the authors to my question about the NK cell adoptive transfer experiments adds very useful context, which I think readers of the article would benefit from. I realise the manuscript is already lengthy, but if there is scope to include Figure R2 in the Supplementary Information file and to include some of the explanatory text in the manuscript then I would encourage the authors to do so.

Line 369 - The k in NK cells requires capitalising

Reviewer #3: (No Response)

PLOS authors have the option to publish the peer review history of their article (what does this mean?). If published, this will include your full peer review and any attached files.

Reviewer #1: No

Reviewer #2: No

Reviewer #3: No

---

## [Editor Report · Acceptance letter]

20 Jul 2023

Dear Dr. Hamon,

We are delighted to inform you that your manuscript, "*Streptococcus pneumoniae* drives specific and lasting Natural Killer cell memory," has been formally accepted for publication in PLOS Pathogens.

Best regards,

Kasturi Haldar

Editor-in-Chief

PLOS Pathogens

orcid.org/0000-0001-5065-158X

Michael Malim

Editor-in-Chief

PLOS Pathogens

orcid.org/0000-0002-7699-2064